# A SHARP KL CONVERGENCE ANALYSIS FOR DIFFUSION MODELS UNDER MINIMAL ASSUMPTIONS

**Nishant Jain & Tong Zhang**
University of Illinois Urbana-Champaign
{nj27,tozhang}@.illinois.edu

## ABSTRACT

Diffusion-based generative models have emerged as highly effective methods for synthesizing high-quality samples. Recent works have focused on analyzing the convergence of their generation process with minimal assumptions, either through reverse SDEs or probability flow ODEs. The best known guarantees, without any smoothness assumptions, for the KL divergence so far achieve a linear dependence on the data dimension $d$ and an inverse quadratic dependence on accuracy level $\varepsilon$. In this work, we present a refined analysis for the standard Exponential Integrator discretization that improves the dependence on $\varepsilon$, at the same time maintaining the linear dependence on $d$. Following recent works on higher order/randomized midpoint discretizations, we model the generation process as a composition of two steps: a reverse ODE step followed by a smaller noising step, which leads to better dependence on step size. We then provide a novel analysis which achieves linear dependence on $d$ for the ODE discretization error without any smoothness assumptions. Specifically, we introduce a general ODE-based counterpart of the stochastic localization argument from Benton et al. (2023) and develop new proof techniques to bound second-order spatial derivatives of the score function – terms that do not arise in previous diffusion analyses and cannot be handled by existing techniques. Leveraging this framework, we prove that $\tilde{O}\left(\frac{d \log^{3/2}(1/\delta)}{\varepsilon}\right)$ steps suffice to approximate the target distribution—corrupted by Gaussian noise of variance $\delta$—to within $O(\varepsilon^2)$ in KL divergence, improving upon the previous best result requiring $\tilde{O}\left(\frac{d \log^2(1/\delta)}{\varepsilon^2}\right)$ steps.

## 1 INTRODUCTION

Recently, diffusion based models have picked up momentum for various use-cases involving generative modelling. They are widely used for image generation (Song & Ermon, 2019; Croitoru et al., 2023; Song et al., 2020a; Nichol et al., 2021; Song et al., 2021; Ho et al., 2020), video generation (Epstein et al., 2023; Chen et al., 2023d), semantic editing (Lugmayr et al., 2022), generating text (Li et al., 2022) or audio signals (Liu et al., 2023), protein design (Gruver et al., 2023; Guo et al., 2024), and many other areas. The success of these diffusion models largely stems from their ability to generate high-quality samples using a denoising mechanism. This is achieved by defining a forward noising process that gradually perturbs data from the target distribution, and learning a *score* function using these noisy observations. New samples are then generated by iteratively simulating the reverse of this process, guided by the learned score function. The forward process can be modelled as a stochastic differential equation (SDE) (Song et al., 2020b), and consequently the generation can be carried out by simulating its reverse-time SDE through discretization. Corresponding to this reverse-time SDE, there also exists a probability flow ordinary differential equation (ODE) Song et al. (2020b), which shares the same marginal distributions at all times. Consequently, two main approaches have emerged for sample generation: simulating the reverse SDE (Song et al., 2020b; Chen et al., 2023c) and simulating this probability flow ODE (Chen et al., 2023b; Lu et al., 2022).

Several works (Chen et al., 2023c;a; Lee et al., 2022; Benton et al., 2024; Li & Yan, 2024) have targeted the theoretical underpinnings behind the working of these diffusion models, under various

assumptions. These studies established polynomial convergence rates with respect to the data dimension $d$, assuming accurate score estimation together with regularity conditions such as smoothness of the score function or bounded support of the data distribution. More recent efforts (Chen et al., 2023a; Li & Yan, 2024; Benton et al., 2024) aim to minimize such assumptions and obtain guarantees just using accuracy assumption for the estimated score. The best existing result (Li & Yan, 2024) shows $O(d/T)$ convergence rate in the total varation (TV) distance for the Denoising Diffusion Probabilistic Model (DDPM) (Ho et al., 2020). A recent work (Benton et al., 2024) also achieved linear dependence on the data dimension for convergence in KL divergence, requiring $\tilde{O}(\frac{d}{\varepsilon^2})$ steps to achieve KL divergence within $\varepsilon^2$ with respect to Gaussian perturbation of the true data distribution. Since TV is bounded by square root of the KL divergence, it is an important issue to investigate whether a better convergence rate is achievable in the KL-divergence. While the linear dependence on $d$ seems satisfactory, the quadratic dependence on $\frac{1}{\varepsilon}$ may not be optimal. In this work, we are interested in investigating the following question:

> *Can we improve the dependence of the KL-divergence on $\varepsilon$ while maintaining linear dependence on the dimension, thereby establishing stronger convergence guarantees for diffusion models?*

To achieve this goal, we explore the line of works (Chen et al., 2023b; Gao & Zhu, 2025; Li et al., 2024b;a) which investigate the probability flow ODE for generation. This perspective is motivated by the observation that we can have better discretization dependence for each interval when analyzing the Wasserstein type error directly using the ODEs (Chen et al., 2023b). However, aggregating and bounding the error across all the intervals just based on this ODE requires additional assumptions either related to the error in divergence (Li et al., 2024b) or the Jacobian (Li & Yan, 2024) of the approximated score. Therefore, Chen et al. (2023b) instead considers smoothness of true and approximate score function at all times to bound the Wasserstein error in each interval using the reverse ODE and adds a noising step utilizing Langevin dynamics to then convert the Wasserstein error to TV. However, this noising via Langevin finally results in a suboptimal dependency on $\varepsilon$. Given the improved dependence on step size the probability flow ODE can offer, we also consider using it but instead of additional assumptions or the Langevin dynamics, we just consider taking a smaller step in the forward (noising) direction. This way we are able to consider the error due to discretization on the reverse ODE and then convert it into KL error via the noise addition with a better dependence on step size for each interval, which can then be aggregated across all the intervals. The combination of the ODE step and a smaller noise step can be interpreted as an alternative simulation of the reverse SDE based generation process. This idea of noise addition along the forward process to convert the Wasserstein type error achieved via ODE-based deterministic step to KL has been used in works targeting second order discretization (Li & Cai, 2024), randomized midpoint analysis under smoothness (Li & Jiao, 2024) and also for convergence analysis for the ODE-based consistency model framework (Jain et al., 2025).

Unlike Chen et al. (2023b), we work in the minimal assumptions scenario similar to Chen et al. (2023a); Benton et al. (2024) and just consider the accurate score estimation assumption. A straightforward way then is to consider the analysis of Li & Cai (2024) and adapt it to the standard DDPM sampler. This improves the dependence on $\varepsilon$ but worsens $d-$dependence leading to a complexity of $\tilde{O}\left(\frac{d^{3/2}}{\varepsilon}\right)$. This is discussed further in section 4.1. Therefore, achieving the desired linear convergence rate for the KL-divergence based on these prior works is non-trivial.

To achieve the linear dependence on $d$ in this setup for our considered ODE step followed by noising path, we take inspiration from Benton et al. (2024) (which considers the reverse SDE and by establishing equivalence to stochastic localization directly picks up a known result from the literature) and investigate additional relations between the score function and its derivative. As discussed in the paper, our analysis along this ODE based path introduces additional challenges: it involves terms containing Laplacian of the score function along with the terms containing both score function and its Jacobian, making it more complicated than the SDE counterpart. By establishing the required novel relations between the score function and its higher order gradient terms, we are able to achieve the linear dependence on data dimension $d$ for the ODE, matching the result of Benton et al. (2024). Due to our improved dependence on discretization step size, this translates into a new

*state-of-the-art* guarantee for the KL convergence: requiring $\tilde{O}(d/\varepsilon)$ iterations to achieve $\varepsilon^2-$KL divergence improving up the previous best result of $\tilde{O}(\frac{d}{\varepsilon^2})$ (Benton et al., 2024). Also, since the TV distance is upper bounded by the square root of KL-divergence, this becomes the *state-of-the-art* convergence guarantee for diffusion models as against the TV convergence guarantee provided in Li & Yan (2024).

## 1.1 RELATED WORK

Here, we provide a review of the recent works targeting diffusion based generation broadly categorized into whether they consider the reverse SDE or the probability flow ODE.

**SDE-Based generation.** The effectiveness of this forward noising and the corresponding denoising process for generation was first majorly advocated by the Diffusion Probabilistic Models (DDPM) framework introduced by Ho et al. (2020), which utilized Gaussian transition kernels for noising and estimated the parameters of the corresponding Gaussian denoising kernels using denoising score matching during training. Going further it was shown that this forward noising in DDPMs can be seen as an SDE (Song et al., 2020b) and the generation process then corresponds to the reverse SDE. Since then there have been various works (Chen et al., 2023c; Li et al., 2023; 2025; Lee et al., 2022) targeting the convergence of this generation process. To advocate for the usability in the real world, some recent works (Chen et al., 2023a; Benton et al., 2024; Li & Yan, 2024) have also targeted setups for the SDE based generation methods requiring minimal assumptions (just the bound on score estimation during training via denoising score matching) and have achieved state-of-the-art convergence guarantees. Specifically, Benton et al. (2024) shows only $O(d/\varepsilon^2)$ steps are required to be $\varepsilon^2$-close in KL w.r.t a Gaussian perturbation of the target distribution. On the other hand, Li & Yan (2024) considers the TV-distance and shows $O(d/\varepsilon)$ steps are required to achieve $\varepsilon-$close TV of the perturbed data distribution.

**ODE based generation.** Song et al. (2020b) highlighted that corresponding to the forward noising process for this diffusion model setup, there also exists a probability flow ODE along side the reverse SDE which shares the same marginal distribution at all times. It also advocated that this Probability Flow ODE can lead to faster sampling using the ODE solvers. Taking inspiration, Song et al. (2020a) then proposed a deterministic counterpart of the DDPM sampler and since then various works have attempted to investigate the convergence of these deterministic samplers (Li et al., 2024a; Gao & Zhu, 2025; Huang et al., 2025; Li et al., 2023; 2024b) under various additional assumptions. The current best result (Li et al., 2024a) achieves a TV distance of $\varepsilon$ (w.r.t. perturbation of the true data distribution) in $O(\frac{d}{\varepsilon})$ steps under score estimation and an additional assumption on the Jacobian of the estimated score. Another work (Li et al., 2024b) requires a weaker assumption on the divergence of the estimated score but achieves sub-optimal results. These works have also argued that under just the score estimation assumption, the TV-distance for these deterministic samplers is lower bounded unlike SDE and thus, such additional assumptions are required. Another line of work is based on the predictor-corrector sampling (Song et al., 2020b; Chen et al., 2023b) which uses an ODE step and addition of small noise using Langevin dynamics for smoothening the trajectory to avoid the error blow-up due to ODE. For this scenario, the convergence can be achieved (Chen et al., 2023b) under standard assumptions on score estimation and the smoothness of the true score as well as the approximated score function, which can be further improved using the randomized midpoint discretization in the predictor step (Gupta et al., 2025). Instead of this langevin step, some recent works (Li & Cai, 2024; Li & Jiao, 2024) have considered the ODE step with second order or randomized midpoint (under smoothness) discretizations and then directly adding noise along the forward process, thereby improving the dependence on $\varepsilon$. In this work, we also take a similar route but for the DDPM sampler and achieve state-of-the-art KL convergence rate under just the score estimation assumption.

## 2 PRELIMINARIES AND SETUP

We now discuss the formulation behind diffusion models in detail, including both ODE and SDE-based generation. Following this, we discuss the assumptions used to achieve the results provided in the next section.

**SDE considered and its discretization.** As discussed previously, diffusion models are based on a forward noising process and the corresponding reverse generation process. The forward process for $d-$dimensional setup can be seen as taking the given samples and gradually corrupting them using the SDE of the following form (Song et al., 2020b):

$$dx(t) = -\mu(x(t), t)dt + g(t)dw_t$$

where $x(0) = y \sim p_{data}$, $x(t) \in \mathbb{R}^d$, $\mu$ and $g$ correspond to the drift and diffusion coefficients, $w_t$ is the $d-$dimensional Brownian motion. Following the popular choice of the OU process, we consider the following SDE:

$$dx(t) = -x(t)dt + \sqrt{2}dw_t$$

The corresponding OU process would be:

$$x(t) = e^{-t}y + \sqrt{1 - e^{-2t}} \cdot \epsilon(t), \qquad \epsilon(t) \sim \mathcal{N}(0, I_d) \tag{1}$$

where $p_t$ denotes the law at time $t$, $x(t) \sim p_t$ and $y \sim p_{data}$. Also, the joint distribution of the random variables generated via this process at time-stamps corresponding to a sequence $\{t_1, .., t_K\}$: $(x_{t_1}, ..., x_{t_K})$ is denoted as $p_{t_1,...,t_K}$. The resulting reverse SDE (Song et al., 2020b) for generation will be:

$$dx(t) = -x(t)dt - 2\nabla \ln p_t(x(t))dt + \sqrt{2}d\bar{w}_t \tag{2}$$

where $\nabla \ln p_t(x(t))$ is referred to as the *score* function and $\bar{w}_t$ is again the Brownian motion. If the forward process is run from time $T$ then initializing from $p_T$ and going along this reverse process for a time $T - t$ will result in the marginal $p_t$. For the corresponding probability flow ODE (Song et al., 2020b), we have the following equation:

$$dx(t) = -x(t)dt - s(t, x(t))dt \tag{3}$$

where we denote $s(t, \cdot) = \nabla \log p_t(\cdot)$. Using the Exponential Integrator discretization (Chen et al., 2023a) where we divide the overall generation time into small intervals and fix the input to the score function for each interval to be the value at the start (from the reverse direction), leads to the following ODE for the interval $[t_{k-1}, t_k]$:

$$dx(t) = -x(t)dt - s(t_k, x_k)dt$$

**Empirical Counterpart.** Practically, we do not have the true score function $s(t, \cdot)$ and instead during training it is approximated via *denoising score matching* (Song et al., 2020a). Denoting that approximated score function as $\hat{s}(t, \cdot)$, we have the following empirical version (discretized and using the approximate score) of the true ODE:

$$d\hat{x}(t) = -\hat{x}(t)dt - \hat{s}(t_k, \hat{x}_k)dt \tag{4}$$

where we denote the law of this empirical process at time $t$ as $\hat{p}_t$. For any particular discretization $\{t_k\}_{k=1}^N$ of the reverse process, this process is usually initialized using a normal distribution $\hat{x}_{t_k} \sim \mathcal{N}(0, I_d)$ and we denote the joint distribution for the true $(x_{t_1}, ..., x_{t_N})$, reverse $(\hat{x}_{t_1}, ...., \hat{x}_{t_N})$ processes as $p_{t_1,...,t_N}$, $\hat{p}_{t_1,...,t_N}$. Also the conditional distribution at $t_{k-1}$ conditioned on $t_k$ is denoted as $p_{t_{k-1}|t_k}$, $\hat{p}_{t_{k-1}|t_k}$ for the true and empirical processes respectively. We now discuss the assumptions used in our theoretical framework.

**Assumptions.** As discussed in the introduction, for our theoretical analysis, we take inspiration from the line of works operating under minimal assumptions (Benton et al., 2024; Chen et al., 2023a; Li & Yan, 2024), and just use the following standard assumptions:

**Assumption 2.1.** *For the discretization sequence $\{t_k\}_{k=1}^{K+1}$ discussed in the next section (and used in the Inference Algorithm 1), the score function estimate $\{\hat{s}(t, \cdot)\}_{1 \leq t \leq T}$ satisfies:*

$$\frac{1}{T} \sum_{k=1}^{K+1} h_k \mathbb{E}_{x \sim p_{t_k}} \left[ \|\hat{s}(t_k, x) - s(t_k, x)\|^2 \right] \leq \varepsilon_{score}^2. \tag{5}$$

where $h_k = t_k - t_{k-1}$ corresponds to the step size of the discretization.

**Assumption 2.2.** The data distribution $p_{data}$ has finite second order moment $\mathbb{E}_{x_0 \sim p_{\text{data}}} \left[ \|x_0\|_2^2 \right] = m_2 < \infty$.

## 2.1 NOTATIONS

As discussed above, $y$ corresponds to the data distribution $p_{data}$, $x(t)$ (with its law denoted by $p_t$ and score function as $s(\cdot)$) corresponds to the forward OU process and $z(t)$ (with its law denoted by $q_t$ and score function as $s_r(\cdot)$) is the variance exploding counterpart of the forward process discussed in Appendix. $\tilde{x}(t)$ corresponds to the discretized version of the true reverse process. $\hat{x}'_k$ denotes the sequence of random variables generated by our algorithm for a discretization sequence $\{t_k\}$ and their law is denoted by $\hat{p}_{t_k}$. The step size $t_k - t_{k-1}$ is denoted as $h_k$. $x_k$ corresponds to the random variables generated by the forward process for this time sequence. $\tilde{x}_{k-1}, \hat{x}_{k-1}$ corresponds to random variables generated by running our proposed scheme (with true, empirical probability flow ODE respectively) for a single interval $[t_{k-1}, t_k]$ starting from $x_k$ at $t_k$. $\tilde{x}_{k-0.5}, \hat{x}_{k-0.5}$ corresponds to the random variable generated by taking two steps along the discretized (Empirical, True respectively) probability flow ODE in reverse direction starting from $x_k$. $x_{k-0.5}$ denotes two steps of true probability flow ODE from $x_k$. $\nabla s(t, x)$ denotes the Jacobian of the score and $\partial_t$ corresponds to the partial derivative w.r.t. time $t$. We further define $\partial_i$ as the partial derivative w.r.t. $i^{th}$ coordinate of the spatial variable $x$ (or $z$ discussed in appendix). It can also be interpreted as $\partial_{x_i}/\partial_{z_i}$. We also define Laplacian operator $\Delta = \sum_{i=1}^{d} \partial_i \partial_i$. The $i^{th}$ element of the score vector $s(\cdot)$ is denoted by $s(\cdot)_i$. $\mathcal{N}(0, I_d)$ denotes the d-dimensional standard Normal distribution. For two terms $P, Q$ $P \lesssim Q$ means there exist an absolute constant $C_1$ such that $P \leq C_1 Q$.

## 3 MAIN RESULTS

As discussed above, using previous works (Chen et al., 2023b) on the probability flow ODE, we can directly analyze the Wasserstein-type error under smoothness conditions by using the Young's and Grönwall's inequality. However, the aggregation leads to blow-up in the error and thus, noise is added via Langevin dynamics based corrector after a small ODE step to instead convert the error into TV distance, which can then be aggregated. Here, we instead take a more simplistic perspective for the diffusion use-case where we consider first taking a step along the reverse ODE to bound the Wasserstein-type error (without any smoothness conditions) and then taking a partial step in the noising (forward) direction to convert this error to KL. As discussed in the introduction, this is inspired from recent works (Jain et al., 2025; Li & Jiao, 2024; Li & Cai, 2024) which target the consistency model setup or randomized midpoint/second order discretization schemes.

We first define a discretization sequence $0 < \delta = t_0 < t_1 < t_2 < ... < t_K < t_{K+1} = T$ for the generation process, where $T$ denotes the total time, initializing it from the standard normal distribution $\mathcal{N}(0, I_d)$. Denoting $h_k = t_k - t_{k-1}$, we provide the inference procedure in Algorithm 1. It is based on the Exponential Integrator discretization of the empirical probability flow ODE (Eq. 4) in the step 4 followed by noise addition along the forward process (Eq. 1) in step 6. The step along the ODE can be used to control the Wasserstein-type error and then the noise addition can convert this into KL. This is discussed further in the next section and in detail (along with the technical lemmas) in Appendix A.1. We denote the generated sequence from our algorithm by random variables $\hat{x}'_k$ (corresponding to time $t_k$) and their law by $\hat{p}_{t_k}$ and the joint distribution for the complete sequence as $\hat{p}_{t_1,...,t_K,t_{K+1}}$. Similarly, corresponding to the sequence generated by the forward process along these time-stamps, we will have the joint distribution as $p_{t_1,...,t_K,t_{K+1}}$. Figure 1 shows this algorithm/generation process.

---

**Algorithm 1** Inference Algorithm for Diffusion Models

---

1: **Given:** Discretizing sequence $\{t_0, t_1, .., t_K, t_{K+1}\}$, $h_k = t_k - t_{k-1}$, $\hat{p}_{t_{K+1}}$ as the normal distribution $\mathcal{N}(0, I_d)$
2: Sample $\hat{x}'_{K+1} \sim \hat{p}_{t_{K+1}}$
3: **for** $k = K+1, K, \ldots, 2$ **do**
4: $\quad \hat{x}'_{k-0.5} = e^{h_k + h_{k-1}} \hat{x}'_k + (e^{h_k + h_{k-1}} - 1)\hat{s}(t_k, \hat{x}'_k)$
5: $\quad$ Sample $\eta_k \sim \mathcal{N}(0, I_d)$
6: $\quad \hat{x}'_{k-1} = e^{-h_{k-1}} \hat{x}'_{k-0.5} + \sqrt{1 - e^{-2h_{k-1}}} \, \eta_k$
7: **end for**
8: **Output** $\hat{x}'_1$

---

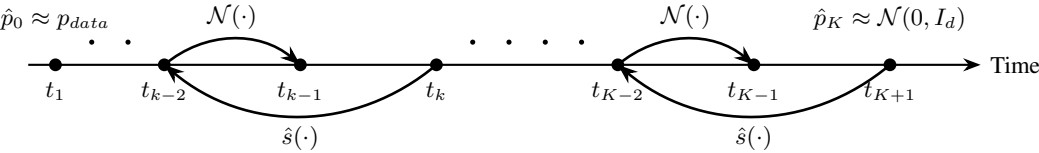

Figure 1: Demonstrating the two updates: (a) along the generation process using $\hat{s}(\cdot)$ and (b) the forward noising process ($\mathcal{N}(\cdot)$), of our proposed scheme.

We now provide the guarantee for the distribution generated by Algorithm 1 in terms of KL divergence w.r.t. perturbation of the true data distribution (law of the forward process at $t_1$): $p_{t_1}$. This corresponds to early stopping (with $t_1 > \delta > 0$), similar Benton et al. (2024); Chen et al. (2023a), as we also avoid smoothness assumptions on the data distribution.

**Theorem 3.1.** *For $T \geq 1$ and $K > d(\log(\frac{1}{\delta}) + T)$, under Assumptions 2.1 and 2.2, consider the generation process in Algorithm 1 with discretization times $0 < \delta = t_0 < t_1 < \cdots < t_{K+1} = T$ defined by the step size rule $h_k = t_k - t_{k-1} = c\min\{1, t_k\}$ for some constant $c > 0$. Then, denoting $\hat{p}_{t_k}$ as the marginal distribution at $t_k$ for this algorithm and $p_t$ as the distribution of the forward process (Eq. 1) at time $t$, we have:*

$$\mathrm{KL}\left(p_{t_1}\big\|\hat{p}_{t_1}\right) \lesssim (d + m_2)e^{-T} + d^2c^3K + T\varepsilon_{score}^2 \qquad (6)$$

*where $x \lesssim y$ means there exists an absolute constant $C$ such that $x \leq Cy$.*

We provide the proof of this theorem in the Appendix (A.5) and discuss a sketch of the complete proof in the next section. The first term corresponds to the error due to initializing the algorithm from $\mathcal{N}(0, I_d)$, second term corresponds to the error due to discretization and the third term is due to error in score estimation (Assumption 2.1). From the definition of $c$, it can be observed that (discussed in the proof as well) $c^3$ should be $O\left(\frac{(\log\frac{1}{\delta}+T)^3}{K^3}\right)$. Also we can observe that $T$ is required to just have a logarithmic dependence on $d$ and thus, the second term corresponding to the discretization error will be $\tilde{O}\left(\frac{d^2}{K^2}\right)$. We formalize this in the following corollary discussing the iteration complexity.

**Corollary 3.2.** *Under assumptions 2.1, 2.2 running Algorithm 1 for the SDE based generation via diffusion models for a total time $T = \log\left(\frac{d+m_2}{\varepsilon_{score}}\right)$ with an exponentially decaying step size sequence $h_k = t_k - t_{k-1} = c\min\{t_k, 1\}$ where $c = \Theta\left(\frac{\log(\frac{1}{\delta})+T}{K}\right)$ achieves a KL-divergence error of $\tilde{O}(\varepsilon_{score}^2)$ with an iteration complexity $K = \Theta\left(\frac{d\left(\log(\frac{1}{\delta})^{3/2}\right)}{\varepsilon_{score}}\right)$, improving upon the previous best complexity of $\Theta\left(\frac{d\log^2(\frac{1}{\delta})}{\varepsilon_{score}^2}\right)$ (Benton et al., 2024).*

## 4 PROOF SKETCH

We now provide a brief sketch of the proof for Theorem 3.1 and the complete details are provided in the Appendix. We begin by first discussing the decomposition of KL divergence into the Wasserstein-type error aggregated in each interval. Since the ODE can result in a better dependence on the discretization step size (Chen et al., 2023b), this serves as the main motivation of our Algorithm 1. Then, we discuss bounding the discretization error along this ODE path in the non-smooth scenario. Finally, we discuss on how the optimal dependence on $d$ can be achieved for this non-smooth setup, leading to state of the art convergence guarantee for the KL divergence.

**KL control for diffusion via Wasserstein-type error.** We can first decompose the KL between between the generation process and the forward process at $t_1$: $\mathrm{KL}\left(p_{t_1}\big\|\hat{p}_{t_1}\right)$ using the data process-

ing inequality and chain rule as follows (Lemma A.2):

$$\mathrm{KL}\left(p_{t_1}\big\|\hat{p}_{t_1}\right) \leq \mathrm{KL}\left(p_{t_{K+1}}\big\|\hat{p}_{t_{K+1}}\right) + \mathbb{E}_{p_{t_1,..,t_{K+1}}}\left[\sum_{k=2}^{K+1}\mathrm{KL}\left(p_{t_{k-1}|t_k}(\cdot|x_k)\big\|\hat{p}_{t_{k-1}|t_k}(\cdot|x_k)\right)\right]$$

where $p_{t_{k-1}|t_k}$ denotes the conditional distribution of the true process at $t_{k-1}$ given $x_k$ at $t_k$ and similarly $\hat{p}_{t_{k-1}|t_k}$ for the generation process. The first term on the RHS is just the initialization error (the error by using the standard normal distribution for initialization as against the distribution of the forward process after time $T$) and can be bounded following previous works (Chen et al., 2023c;a) as $(d+m_2)e^{-T}$. The second term denotes the summation of the KL error aggregated in each interval $[t_{k-1}, t_k]$ when the true and the generation process start from the same point ($x_k$). Now, to calculate this term, we will consider the following update for the interval $[t_{k-1}, t_k]$ starting from $x_k$ using the empirical ODE (Eq. 4) and noise:

$$\hat{x}_{k-0.5} = e^{h_k+h_{k-1}}x_k + (e^{h_k+h_{k-1}}-1)\hat{s}(t_k, x_k) \tag{7}$$

$$\hat{x}_{k-1} = e^{-h_{k-1}}\hat{x}_{k-0.5} + \sqrt{1-e^{-2h_{k-1}}}\epsilon_k, \qquad \epsilon_k \sim N(0, I). \tag{8}$$

Based on this, the $\mathrm{KL}\left(p_{t_{k-1}|t_k}(\cdot|x_k)\big\|\hat{p}_{t_{k-1}|t_k}(\cdot|x_k)\right)$ term can be written as (Lemma A.1):

$$\mathrm{KL}\left(p_{t_{k-1}|t_k}(\cdot|x_k)\big\|\hat{p}_{t_{k-1}|t_k}(\cdot|x_k)\right) = e^{-2h_{k-1}}\frac{\|x_{k-0.5} - \hat{x}_{k-0.5}\|_2^2}{2(1-e^{-2h_{k-1}})}$$

where $x_{k-0.5}$ denotes the true reverse process at time $t_k - h_k - h_{k-1}$. This Wasserstein-type error to KL conversion and then aggregation is inspired from the recent works (Jain et al., 2025; Li & Cai, 2024; Li & Jiao, 2024). We now discuss on how the expected value of the $\|x_{k-0.5} - \hat{x}_{k-0.5}\|_2^2$ term in the RHS of the last equation can be bounded to finally bound the expression obtained after applying the chain rule.

## 4.1 BOUNDING $\mathbb{E}[\|x_{k-0.5} - \hat{x}_{k-0.5}\|^2]$

For error control of this term, we define an additional process for the interval $[t_{k-1}, t_k]$ (starting from $x_k$ and governed by Exponential Integrator discretization of the true probability flow ODE in Eq. 3): $\tilde{x}_k$:

$$\tilde{x}_{k-0.5} = e^{h_k+h_{k-1}}x_k + (e^{h_k+h_{k-1}}-1)s(t_k, x_k) \tag{9}$$

$$\tilde{x}_{k-1} = e^{-h_{k-1}}\tilde{x}_{k-0.5} + \sqrt{1-e^{-2h_{k-1}}}\epsilon_k, \quad \epsilon_k \sim N(0, I). \tag{10}$$

Now, we decompose the target term corresponding to our scheme $\mathbb{E}\|\hat{x}_{k-0.5} - x_{k-0.5}\|_2^2$ for each interval as follows:

$$\sqrt{\mathbb{E}[\|x_{k-0.5} - \hat{x}_{k-0.5}\|_2^2]} \leq \underbrace{\sqrt{\mathbb{E}[\|x_{k-0.5} - \tilde{x}_{k-0.5}\|_2^2]}}_{\mathrm{T}_d} + \underbrace{\sqrt{\mathbb{E}[\|\tilde{x}_{k-0.5} - \hat{x}_{k-0.5}\|_2^2]}}_{\mathrm{T}_s} \tag{11}$$

where $\mathrm{T}_s$ is the error due to using the approximate score function $\hat{s}$ and $\mathrm{T}_d$ is the error due to the discretization of true process. The score estimation error term can be written as $(e^{h_k+h_{k-1}}-1)^2\mathbb{E}[\|s(t_k, x_k) - \hat{s}(t_k, x_k)\|_2^2]$ (Lemma A.3) and aggregated across all the intervals can be bounded as $O(h_k\varepsilon_{score}^2)$ using Assumption 2.1 (further discussed in the proof of Theorem 3.1 in Section A.5).

To bound the discretization error, we first define a rescaled version of the original process as $z(t) = e^t x(t)$ (Section A.3) with the law at time denoted by $q_t$, score function denoted as $s_r(t, z(t))$. This is done to simplify the calculations in the analysis. Now, using the ODE path of Eq. 14 and the Integral Remainder form of the Taylor Expansion, we bound this discretization error (Lemma A.4):

$$\mathbb{E}\left[\|z_{k-0.5} - \tilde{z}_{k-0.5}\|_2^2\right] \leq \frac{1}{2}(h_k+h_{k-1})^3 \int_{t_{k-2}}^{t_k} e^{4t}\mathbb{E}\left[\|s_r'(t, z(t))\|_2^2\right] dt$$

where the derivative of the score $s_r'(t, z(t))$ can be calculated as $s_r'(t, z(t)) = \frac{d}{dt}s_r(t, z(t)) = \frac{\partial s_r(t,z)}{\partial t} + \frac{\partial s_r(t,z)}{\partial z}\frac{dz(t)}{dt}\Big|_{z=z(t)}$. This is different from previous works (Benton et al., 2024; Chen et al.,

2023a) which instead consider the reverse SDE in Eq. 2 and thereby incur the discretization error contribution for each interval in the overall KL divergence as: $\int_{t_{k-1}}^{t_k} \mathbb{E}\left[\|s(t_k, x_k) - s(t, x(t))\|^2\right] dt$, bounded using the Jacobian of the score. This results in a sub-optimal dependence on $h_k$ ($O(h_k^2)$). Specifically, the best bound is achieved in Benton et al. (2024) which directly expresses the derivative of $\mathbb{E}\left[\|s(t_k, x_k) - s(t, x(t))\|^2\right]$ term w.r.t. $t$ in terms of $\mathbb{E}_{p_t}\left[\|\nabla s(t, x)\|_F^2\right]$ and then integrates the bound on this, resulting in the optimal linear dependence on $d$ and a sub-optimal $O(h_k^2)$ dependence on the step size.

Now, we discuss in detail on how to bound $\int_{t_{k-2}}^{t_k} \mathbb{E}\left[\|s_r'(t, z(t))\|_2^2\right] dt$ following Eq 14. Using the trick proposed in Chen et al. (2023a), we can write $z(t)$ as a Gaussian perturbation in $y \sim p_{data}$, thereby rewriting score function at time $t$: $s_r(t, z)$ as $\mathbb{E}_{y|z}\left[\frac{y-z}{1-e^{-2t}}\right]$ (Lemma A.5). A straightforward option is to then calculate the Jacobian ($\nabla s_r(t, z)$, partial derivative w.r.t. time ($\partial_t s_r(t, z)$) and then substitute in the expression of $s_r'(t, z(t))$ to get an expression for $\mathbb{E}\left[\|s_r'(t, z(t))\|_2^2\right]$. This can then be upper bounded using the fact that $\frac{y-z}{\sqrt{e^{2t}-1}} \sim \mathcal{N}(0, I_d)$ (Chen et al., 2023a) (also discussed in Lemma A.8). This is just the adaptation of the calculations proposed in Li & Cai (2024) for the considered DDPM sampler and leads to a better dependence on $h_k$ than Benton et al. (2024) but will lead to $d^3$ dependence of our target term $\left(\int_{t_{k-2}}^{t_k} \mathbb{E}\left[\|s'(t, x(t))\|_2^2\right] dt\right)$ and thus, a $d^{3/2}$ dependence for KL $\left(p_{t_1} \| \hat{p}_{t_1}\right)$, which is worse than the $d$-dependence achieved for KL in Benton et al. (2024).

### 4.1.1 ACHIEVING THE OPTIMAL $d$-DEPENDENCE FOR ODE

Lemma A.8 shows that $\mathbb{E}\left[\|s_r(t, z)\|^2\right]$ can be bounded as $O(\frac{d}{e^{2t}-1})$ as against $O(\frac{d^2}{(e^{2t}-1)^2})$ for $\mathbb{E}\left[\|\nabla s_r(t, z)\|_F^2\right]$. Therefore to have the linear $d-$dependence, we take inspiration from Benton et al. (2024) which first establishes the equivalence of reverse SDE based to Stochastic Localization and then exploits a well known result from the Stochastic Localization literature (Lemma 1 in the paper). Since we are considering the ODE path, instead of directly utilising such result, we begin by first establishing $\frac{d}{dt}\mathbb{E}_{q_t}[\|s_r(t, z)\|^2] = -2e^{2t}\mathbb{E}_{q_t}[\|\nabla s_r(t, z)\|_F^2]$ in Lemma A.11. Given our target term for the discretization error $\left(\int_{t_{k-2}}^{t_k} \mathbb{E}_{q_t}\left[\|s_r'(t, z(t))\|_2^2\right] dt\right)$ depends on the integral (w.r.t. time) of the Jacobian term, using $\frac{d}{dt}\mathbb{E}_{q_t}[\|s_r(t, z)\|^2]$ can improve the $d^2$ contribution from this term to $d$. This serves as the motivation for the remaining sketch.

As discussed above, since Benton et al. (2024) involves the reverse SDE, it just requires bounding $\mathbb{E}_{q_t}[\|\nabla s_r(t, z)\|_F^2]$ term. However for our considered probability flow ODE path we need to bound the overall derivative term: $\mathbb{E}\left[\|s_r'(t, z)\|^2\right]$ which includes partial derivative w.r.t. time $\partial_t s_r(t, z)$ making the analysis much more complicated which is discussed next.

We first convert the time-derivative to spatial derivatives using the the Fokker-Planck equation counterpart for the score function (Lemma A.9):

$$\partial_t s_r(t, z) = e^{2t}\Delta s_r(t, z) + 2e^{2t}\nabla s_r(t, z)^\top s_r(t, z)$$

where recall from Section 2.1 that $\Delta$ denotes the Laplacian of the score $s_r$. This, results in the overall derivative term being represented only in terms of spatial derivative as follows (Lemma A.10):

$$\mathbb{E}_{q_t}\left[\|s_r'(t, z)\|^2\right] = e^{4t}\mathbb{E}_{q_t}\left[\|\Delta s_r(t, z)\|_2^2 + \|\nabla s_r(t, z)^\top s_r(t, z)\|_2^2\right]$$
$$+ \mathbb{E}_{q_t}\left[(\Delta s_r(t, z))^\top \left(\nabla s_r(t, z)^\top s_r(t, z)\right)\right] \tag{12}$$

Since this overall derivative involves a term containing both score, its Jacobian and a term containing the Laplacian of the score, bounding this involves more complex analysis as compared to the

SDE scenario in Benton et al. (2024). Now, based on the motivation discussed above to achieve optimal $d$ dependence by expressing $\frac{d}{dt}\mathbb{E}_{q_t}[\|s_r(t,z)\|^2] = -2e^{2t}\mathbb{E}_{q_t}[\|\nabla s_r(t,z)\|_F^2]$, we further establish similar relations of the RHS terms in Eq. 12. For the term comprising both $s_r$ and $\nabla s_r$ in Eq. 12, we provide the generalized version of Lemma A.11 which considers general power $m$ in $\frac{d}{dt}\mathbb{E}_{q_t}[\|s_r(t,z)\|_2^m]$ and a term of the form $\mathbb{E}\left[\|s_r(t,z)\|_2^{m-2}\|\nabla s_r(t,z)\|_F^2\right]$ (Lemma A.12):

$$e^{-2t}\frac{d}{dt}\mathbb{E}_{q_t}[\|s_r(t,z)\|_2^m] = -m\mathbb{E}_{q_t}[\|s_r(t,z)\|_2^{m-2}\|\nabla s_r(t,z)\|_F^2]$$
$$-\frac{m(m-2)}{4}\mathbb{E}_{q_t}\left[\|s_r(t,z)\|_2^{m-4}\left\|\left(\nabla\|s_r(t,z)\|_2^2\right)\right\|_2^2\right]$$

We then utilise this equation to first write the second term in the RHS of our main Eq. 12 in terms of $\frac{d}{dt}\mathbb{E}_{q_t}[\|s_r(t,z)\|_2^m]$. To target the first term, we establish another novel relation by starting with the term $\frac{d}{dt}\mathbb{E}_{q_t}[\|\nabla s_r(t,z)\|_2^2]$ and expressing it in terms of $\int \Delta q_t(z)\|\nabla s_r(t,z)\|_F^2 dt$, $\mathbb{E}_{q_t}[\|\Delta s_r(t,z)\|^2]$ and $\mathbb{E}_{q_t}[\|\nabla\|s_r(t,z)\|_2^2\|_2^2]$. Then, we rearrange and express $\mathbb{E}_{q_t}[\|\Delta s_r(t,z)\|^2]$ in terms of $\int \Delta q_t(z)\|\nabla s_r(t,z)\|_F^2 dt$, $\mathbb{E}_{q_t}\left[\|\nabla\|s_r(t,z)\|_2^2\|_2^2\right]$ and $\frac{d}{dt}\mathbb{E}_{q_t}\left[\|\nabla s_r(t,z)\|^2\right]$ (Lemma A.16). We bound the term $\int \Delta q_t(z)\|\nabla s_r(t,z)\|_F^2 dt$ as follows (Lemma A.15, $C_d$ present in the lemma statement is $O(1)$ since $C_d \leq 12$ for $d \geq 10$):

$$\int \Delta q_t(z)\|\nabla s_r(t,z)\|_F^2 dt \lesssim \frac{d^2}{(e^{2t}-1)^3} - \frac{e^{-2t}d}{(e^{2t}-1)}\frac{d}{dt}\mathbb{E}_{q_t}[\|s_r(t,z)\|^2]$$

leading to an overall bound on the $\mathbb{E}_{q_t}[\|\Delta s_r(t,z)\|^2]$ as (Lemma A.16):

$$\mathbb{E}_{q_t}\left[\|\Delta s_r(t,z)\|_2^2\right] \lesssim \frac{d^2}{(e^{2t}-1)^3} - \frac{de^{-2t}}{(e^{2t}-1)}\frac{d}{dt}\mathbb{E}_{q_t}[\|s_r(t,z)\|^2]$$
$$- e^{-2t}\left(\frac{d}{dt}\mathbb{E}_{q_t}\left[\|\nabla s_r(t,z)\|_F^2\right] + \frac{d}{dt}\mathbb{E}_{q_t}\left[\|s_r(t,z)\|_2^4\right]\right)$$

Finally, this leads to the following bound on $\mathbb{E}_{q_t}[\|s_r'(t,z)\|_2^2]$ (Lemma A.17):

$$\mathbb{E}_{q_t}[\|s_r'(t,z)\|_2^2] \lesssim \frac{d^2 e^{4t}}{(e^{2t}-1)^3} - \frac{e^{2t}d}{(e^{2t}-1)}\frac{d}{dt}\mathbb{E}_{q_t}[\|s_r(t,z)\|^2]$$
$$- e^{2t}\left(\frac{d}{dt}\mathbb{E}_{q_t}\left[\|\nabla s_r(t,z)\|_F^2\right] + \frac{d}{dt}\mathbb{E}_{q_t}[\|s_r(t,z)\|^4]\right)$$

Integrating this and summing up across all the intervals, choosing $h_k = c\min\{t_k, 1\}$ following the previous works (Chen et al., 2023a; Benton et al., 2024) and scaling back to $\tilde{x}(t)$ along with accounting for the score estimation error and the initialization error leads to the following final expression for KL $\left(p_{t_1}\big\|\hat{p}_{t_1}\right)$ (section A.5, refer to the analysis there for more details):

$$\text{KL}\left(p_{t_1}\big\|\hat{p}_{t_1}\right) \lesssim (d+m_2)e^{-T} + d^2c^3K + T\varepsilon_{score}^2$$

where due to the exponentially decaying step size $c \lesssim \frac{\log(\frac{1}{\delta})+T}{K}$ which results in $K = \Theta\left(\frac{d\log^{3/2}(\frac{T}{\delta})}{\varepsilon}\right)$ to achieve $\tilde{O}(\varepsilon^2)$ KL $\left(p_{t_1}\big\|\hat{p}_{t_1}\right)$ error.

## 5 CONCLUSION

In this work we provided an improved analysis for generation process of the diffusion models under just the $L^2$-accurate score estimation and finite second moment of the data distribution assumption. We showed that by modelling the SDE based generation process as an ODE step followed by noising and thereby targeting the discretization error along this ODE path can lead to better dependence on the step size. We also introduced a novel analysis framework for this ODE path which expresses the overall derivative of the score function in terms of spatial derivatives and establishes relations between the score and its first, second order spatial derivatives. This resulted in achieving linear dependence on $d$ for the considered ODE path, leading to a new *state-of-the-art* convergence guarantee for KL divergence. Since KL upper bounds the square of the TV-distance by Pinsker's inequality, our result also provides a stronger guarantee than the best existing rate for the TV convergence achieved in Li & Yan (2024). An interesting future direction can be to investigate if the dependence on the step size can be improved further when considering this ODE step followed by noising framework, thereby enhancing the dependence on $\varepsilon$ and achieving faster convergence.

ACKNOWLEDGMENTS

This material is based upon work supported partially by NSF under Grant No. 2416897, Grant No. 2505932, and by ORN under Grant No. N000142512318. Any opinions, findings, and conclusions or recommendations expressed in this materi al are those of the author(s) and do not necessarily reflect the views of the funding agencies.

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

# A   PROOF OF THEOREM 3.1

## A.1   BOUNDING $\mathrm{KL}\left(p_{t_1}\middle\|\hat{p}_{t_1}\right)$ AS AGGREGATION OF $\mathbb{E}[\|x_{k-0.5} - \hat{x}_{k-0.5}\|_2^2]$ FOR EACH INTERVAL

We now discuss two lemmas: a) The first one converts Wasserstein type error between the empirical and true process to KL for each interval and the second one aggregates the KL across all the intervals.

**Lemma A.1.** *Denoting $\hat{p}_{t_{k-1}|t_k}$ be the conditional probability of $\hat{x}_{k-1}$ given $\hat{x}_k$, and let $p_{t_{k-1}|t_k}$ be the conditional probability of $x_{k-1}$ given $x_k$ using two steps of ODE and one step of noise similar to our algorithm. Then, for the updates in Eq. 7, Eq. 8 (the updates of our Algorithm 1 for each interval given the same starting point for both is the true process at $t_k$: $x_k$), we have:*

$$\mathrm{KL}\left(p_{t_{k-1}|t_k}(\cdot|x_k)\middle\|\hat{p}_{t_{k-1}|t_k}(\cdot|x_k)\right) = e^{-2h_{k-1}}\frac{\|x_{k-0.5} - \hat{x}_{k-0.5}\|_2^2}{2(1 - e^{-2h_{k-1}})}$$

*where we recall that $h_k = t_k - t_{k-1}$ denotes the step size, $x_{k-0.5}$ corresponds to two steps of Probability Flow ODE from $x_k$ and thus, the law is same as of the forward process at time $t_k - h_k - h_{k-1}$.*

*Proof.* For this, we know that from Algorithm 1 that the conditional $\hat{p}_{t_{k-1}|t_k}(\cdot|x_k)$ for the generation process is the following Gaussian:

$$\hat{p}_{t_{k-1}|t_k}(\cdot|x_k) \sim \mathcal{N}\left(e^{-h_{k-1}}\hat{x}_{k-0.5}, \left(1 - e^{-2h_{k-1}}\right)I_d\right)$$

where $I_d$ is the d-dimensional identity matrix. Similarly, for the true process we can just write:

$$p_{t_{k-1}|t_k}(\cdot|x_k) \sim \mathcal{N}\left(e^{-h_{k-1}}x_{k-0.5}, \left(1 - e^{-2h_{k-1}}\right)I_d\right)$$

Now, since the covariance matrices are same for both, we can just use the following formulae for calculating KL between two Gaussians with different means but same variance:

$$\mathrm{KL}\left(p_{t_{k-1}|t_k}(\cdot|x_k)\middle\|\hat{p}_{t_{k-1}|t_k}(\cdot|x_k)\right) = \frac{1}{2}(\mu_1 - \mu_2)^\top \Sigma^{-1}(\mu_1 - \mu_2)$$

where $\mu_1$, $\mu_2$ corresponds to the mean of the two distributions and $\Sigma$ corresponds to their covariance. For this case, we have:

$$\mu_1 = e^{-h_{k-1}}\hat{x}_{k-0.5}$$
$$\mu_2 = e^{-h_{k-1}}x_{k-0.5}$$
$$\Sigma = \left(1 - e^{-2h_{k-1}}\right)I_d$$

Merely substituting these values in the KL formulae will lead to the desired term.

$\square$

For KL-aggregation, we have the following lemma:

**Lemma A.2.** *For the discretization sequence $t_1, ..., t_{K+1}$ and the law corresponding to the generation process in Algorithm 1, we will have (where $p_{t_k}$ denotes the law of true process at time $t_k$):*

$$\mathrm{KL}\left(p_{t_1}\middle\|\hat{p}_{t_1}\right) \leq \mathrm{KL}\left(p_{t_1,t_2,...,t_K,t_{K+1}}\middle\|\hat{p}_{t_1,t_2,...,t_K,t_{K+1}}\right)$$
$$= \mathrm{KL}\left(p_{t_{K+1}}\middle\|\hat{p}_{t_{K+1}}\right) + \mathbb{E}_{p_{t_1,...,t_K,t_{K+1}}}\left[\sum_{k=2}^{K+1}\mathrm{KL}\left(p_{t_{k-1}|t_k}(\cdot|x_k)\middle\|\hat{p}_{t_{k-1}|t_k}(\cdot|x_k)\right)\right]$$

*Proof.* The first inequality is just the data processing inequality and second equation is the chain rule for KL. $\square$

## A.2 ANALYSING $\mathbb{E}[\|x_{k-0.5} - \hat{x}_{k-0.5}\|_2^2]$

We begin by first decomposing the term into a discretization error component and the error due to using the estimated score instead of true score.

**Lemma A.3.** *For the sequence $\hat{x}_{k-0.5}$ generated by Eq. 7, we have:*

$$\mathbb{E}[\|x_{k-0.5} - \hat{x}_{k-0.5}\|_2^2] \leq 2\mathbb{E}[\|x_{k-0.5} - \tilde{x}_{k-0.5}\|_2^2] + 2(e^{h_k+h_{k-1}} - 1)^2\mathbb{E}[\|s(t_k, x_k) - \hat{s}(t_k, x_k)\|_2^2]$$

*where we recall $x_{k-0.5}$ corresponds to two steps of true probability flow ODE from $x_k$ and $\tilde{x}_{k-0.5}$ corresponds to the discretized true process update defined in Eq. 9. The first term in the RHS corresponds to the discretization error ($T_d$) and the second term is the score estimation error ($T_s$).*

*Proof.* We can just bound the LHS as follows:

$$\sqrt{\mathbb{E}[\|x_{k-0.5} - \hat{x}_{k-0.5}\|_2^2]} \leq \underbrace{\sqrt{\mathbb{E}[\|x_{k-0.5} - \tilde{x}_{k-0.5}\|_2^2]}}_{T_{\text{dis}}} + \underbrace{\sqrt{\mathbb{E}[\|\tilde{x}_{k-0.5} - \hat{x}_{k-0.5}\|_2^2]}}_{T_{\text{est}}}$$

where, as discussed in the lemma, $\tilde{x}_{k-0.5}$ is defined in Eq. 9. Squaring both sides and using $2ab \leq a^2 + b^2$, we will have:

$$\mathbb{E}[\|x_{k-0.5} - \hat{x}_{k-0.5}\|_2^2] \leq 2\mathbb{E}[\|x_{k-0.5} - \tilde{x}_{k-0.5}\|_2^2] + 2\mathbb{E}[\|\tilde{x}_{k-0.5} - \hat{x}_{k-0.5}\|_2^2]$$

**Bounding $T_{\text{est}}$.** Now, utilizing the Eq. 7, Eq. 9, we have:

$$\mathbb{E}[\|\tilde{x}_{k-0.5} - \hat{x}_{k-0.5}\|_2^2] = \mathbb{E}\|(e^{h_k+h_{k-1}} - 1)\left(s(t_k, x_k) - \hat{s}(t_k, x_k)\right)\|_2^2$$
$$= (e^{h_k+h_{k-1}} - 1)^2\mathbb{E}[\|s(t_k, x_k) - \hat{s}(t_k, x_k)\|_2^2]$$

$\square$

We discuss the analysis (and eventually bounding it) of the discretization error term $T_d$ in the subsequent subsections.

## A.3 ANALYSING THE DISCRETIZATION ERROR ALONG THE ODE PATH

**Considering a rescaled process.** We consider a rescaled version of the original OU process (Eq. 1) as $z(t) = e^t x(t)$, leading to:

$$z(t) = y + \sqrt{e^{2t} - 1} \cdot \eta; \qquad \eta \sim \mathcal{N}(0, I_d) \tag{13}$$

where $y$ corresponds to the data distribution: $z_0 = x_0 = y \sim p_{data}$ with the corresponding forward SDE being:

$$dz(t) = x(t)de^t + e^t dx(t) = x(t)e^t dt + e^t\left(-x(t)dt + \sqrt{2}d\boldsymbol{w}_t\right) = \sqrt{2}e^t d\boldsymbol{w}_t$$

We denote the law of this process at time $t$ by $q_t(\cdot)$ where $q_t$ is just a pushforward of $p_t(\cdot)$. Also, we denote the score function of this rescaled process as $s_r(t, \cdot)$ where we will have $s_r(t, z(t)) = e^{-t}s(t, e^{-t}z(t))$. The probability flow ODE becomes:

$$dz(t) = -e^{2t}s_r(t, z(t))dt \tag{14}$$

Using the Exponential Integrator discretization for a given interval $[t_{k-1}, t_k]$, here also, we define $\tilde{z}_{k-0.5}$ for this interval when starting from $z_k$:

$$\tilde{z}_{k-0.5} = z_k + \frac{1}{2}e^{2t_{k-2}}(e^{2(h_k+h_{k-1})} - 1)s_r(t_k, z_k) \tag{15}$$

where $t_{k-2} = t_k - h_k - h_{k-1}$. Now, we have the following lemma for bounding $\mathbb{E}\left[\|z_{k-0.5} - \tilde{z}_{k-0.5}\|_2^2\right]$ where $z_{k-0.5}$ is two steps of true probability flow ODE Eq. 14 from $z_k$.

**Lemma A.4.** *For the $\tilde{z}_{k-0.5}$ defined in Eq. 15, we have:*

$$\mathbb{E}\left[\|z_{k-0.5} - \tilde{z}_{k-0.5}\|_2^2\right] \leq \frac{1}{2}(h_k + h_{k-1})^3 \int_{t_{k-2}}^{t_k} e^{4t}\mathbb{E}\left[\|s_r'(t, z(t))\|_2^2\right] dt$$

*where $s_r'(t, z(t))$ is the derivative of $s_r(t, z(t))$ w.r.t. $t$ and can be calculated using:*

$$s_r'(t, z(t)) = \frac{\partial s_r(t, z)}{\partial t} + \frac{\partial s_r(t, z)}{\partial z}\frac{dz(t)}{dt}\bigg|_{z=z(t)}$$

*Proof.* Since, we have used the Exponential Integrator discretization, the ODEs for the interval $[t_{k-2}, t_k]$ corresponding to $z_k, \tilde{z}_k$ (given $\tilde{z}_k = z_k$) are:

$$dz(t) = -e^{2t}s_r(t, z(t))dt \qquad d\tilde{z}(t) = -e^{2t}s_r(t_k, z_k)dt$$

Therefore, we have:

$$\tilde{z}_{k-0.5} - z_{k-0.5} = \int_{t_k}^{t_{k-2}} d(\tilde{z}(t) - z(t)) = \int_{t_k}^{t_{k-2}} e^{2t}\left(s_r(t_k, z_k) - s_r(t, z(t))\right) dt$$

$$= \int_{t_k}^{t_{k-2}} e^{2t}\left(s_r(t_k, z_k) - \left(\underbrace{s_r(t_k, z_k) + \int_{t_k}^t s_r'(u, z_u) du}_{\text{Taylor's Integral Remainder}}\right)\right) dt$$

where in the last step we have just used the Taylor's Integral Remainder form for the score function. Using this, we will have:

$$\mathbb{E}[\|\tilde{z}_{k-0.5} - z_{k-0.5}\|_2^2] = \mathbb{E}\left[\left\|\int_{t_k}^{t_{k-2}} dt \int_{t_k}^t e^{2u}s_r'(u, z(u))du\right\|_2^2\right]$$

$$\leq \mathbb{E}\left[(h_k + h_{k-1})\int_{t_{k-2}}^{t_k} dt \left\|\int_{t_k}^t e^{2u}s_r'(u, z(u))du\right\|_2^2\right]$$

$$\leq \mathbb{E}\left[(h_k + h_{k-1})\int_{t_{k-2}}^{t_k} (t_k - t)\int_t^{t_k} \left\|e^{2u}s_r'(u, z(u))\right\|_2^2 dudt\right]$$

$$= (h_k + h_{k-1})\int_{t_{k-2}}^{t_k} (t_k - t)dt \int_t^{t_k} \mathbb{E}\left[\left\|e^{2u}s_r'(u, z(u))\right\|_2^2\right] du$$

$$= (h_k + h_{k-1})\int_{t_{k-2}}^{t_k} e^{4u}\mathbb{E}\left[\|s_r'(u, z(u))\|_2^2\right] du \int_{t_{k-2}}^u (t_k - t)dt$$

$$\leq \frac{(h_k + h_{k-1})^3}{2}\int_{t_{k-2}}^{t_k} e^{4u}\mathbb{E}\left[\|s_r'(u, z(u))\|_2^2\right] du$$

$\square$

We now calculate and bound the spatial gradient since the partial gradient w.r.t. time can be written in terms of the spatial gradient using the Fokker Planck Equation (FPE).

**Calculating the Jacobian $\nabla s_r(t, z(t))$ for this rescaled process:** We can now observe the following for this rescaled process:

$$q_t(z) = \int q_t(z|y)p_{data}(y)dy \propto \int e^{-\frac{\|z-y\|^2}{2(e^{2t}-1)}} p_{data}(y)dy \tag{16}$$

which takes us to the following formulation of the score function:

**Lemma A.5.** *For the rescaled process in Eq. 13, we have:*

$$s_r(t, z) = \mathbb{E}_{y|z}\left[\frac{y - z}{(e^{2t} - 1)}\right]$$

*Proof.* As discussed in Eq. 16, we can just write the score function as:

$$s_r(t, z) = \nabla \log q_t(z) = \frac{\nabla q_t(z)}{q_t(z)} = \frac{\nabla \int e^{-\frac{\|z-y\|^2}{2(e^{2t}-1)}} p_{data}(y)dy}{\int e^{-\frac{\|z-y\|^2}{2(e^{2t}-1)}} p_{data}(y)dy} = \frac{\int \frac{y-z}{(e^{2t}-1)} e^{-\frac{\|z-y\|^2}{2(e^{2t}-1)}} p_{data}(y)dy}{\int e^{-\frac{\|z-y\|^2}{2(e^{2t}-1)}} p_{data}(y)dy}$$

$$= \int P(y|z)\frac{y - z}{(e^{2t} - 1)}dy$$

$$= \mathbb{E}_{y|z}\frac{y - z}{(e^{2t} - 1)}$$

where $P(y|z) = \frac{P(y,z)}{\int P(y,z)dy}$ and $P(y,z) = e^{-\frac{\|z-y\|^2}{2(e^{2t}-1)}} p_{data}(y)$. $\qquad\square$

**Lemma A.6.** *Jacobian of score. We have the following expression for the Jacobian of the score $\nabla s_r(t, z)$ for the rescaled process $z(t)$:*

$$\nabla s_r(t, z) = \mathrm{Var}_{y|z}\left[\frac{y - z}{e^{2t} - 1}\right] - \frac{I_d}{e^{2t} - 1}$$

*where* $\mathrm{Var}$ *denotes the covariance matrix.*

*Proof.* We begin by calculating (the gradient of P is w.r.t. second variable $z$ in this Lemma),

$$\nabla P(y|z) = \frac{\nabla P(y, z) \cdot \int P(y, z)dy - P(y, z) \cdot \int \nabla P(y, z)dy}{\left(\int P(z, y)dy\right)^2}$$

From the calculations in last Lemma (A.5), we have:

$$\nabla P(y, z) = \frac{y - z}{e^{2t} - 1}P(y, z)$$

and therefore:

$$\nabla P(y|z) = \frac{y - z}{e^{2t} - 1} \cdot \frac{P(y, z)}{\int P(y, z)dy} - \frac{P(y, z)}{\int P(y, z)dy} \cdot \frac{\int \frac{y-z}{e^{2t}-1}P(y, z)dy}{\int P(y, z)dy}$$

$$= P(y|z)\left(\frac{y - z}{e^{2t} - 1} - \mathbb{E}_{y|z}\left[\frac{y - z}{e^{2t} - 1}\right]\right)$$

Thus, we can calculate the $\nabla s_r(t, z)$ as follows:

$$\nabla s_r(t, z) = \int \frac{y - z}{e^{2t} - 1}\nabla P(y|z)^\top dy - \frac{I_d}{e^{2t} - 1}\int P(y|z)dy$$

$$= \int P(y|z)\frac{y - z}{e^{2t} - 1}\left(\frac{y - z}{e^{2t} - 1} - \mathbb{E}_{y|z}\left[\frac{y - z}{e^{2t} - 1}\right]\right)^\top dy - \frac{I_d}{e^{2t} - 1}$$

$$= \mathbb{E}_{y|z}\left[\frac{y - z}{e^{2t} - 1}\left(\frac{y - z}{e^{2t} - 1} - \mathbb{E}_{y|z}\left[\frac{y - z}{e^{2t} - 1}\right]\right)^\top\right] - \frac{I_d}{e^{2t} - 1}$$

$$= \mathrm{Var}_{y|z}\left[\frac{y - z}{e^{2t} - 1}\right] - \frac{I_d}{e^{2t} - 1}$$

$\qquad\square$

**Bounding** $\mathbb{E}_{q_t}\left[\|s_r(t, z)\|^p\right]$, $\mathbb{E}_{q_t}\left[\|\nabla s_r(t, z)\|^2\right]$ **and other spatial gradient terms.** Since, we know that $\frac{y - z(t)}{\sqrt{e^{2t}-1}} = \epsilon \sim \mathcal{N}(0, I_d)$, we first provide a helper lemma to bound the moment of the multivariate Gaussian distribution. Then using that and the formulae for score function, Jacobian provided in Lemma A.5, we bound $\mathbb{E}\left[\|s_r(t, z)\|^p\right]$, $\mathbb{E}\left[\|\nabla s_r(t, z)\|_F^2\right]$ for a general $p$.

**Lemma A.7.** *Gaussian Moment. We have the following result for the Gaussian random variable $\eta \sim \mathcal{N}(0, I_d)$:*

$$\mathbb{E}\left[\|\eta\eta^\top\|_F^p\right] = \mathbb{E}\left[\|\eta\|_2^{2p}\right] \leq (d + 2p)^p$$

*Proof.* We will have:

$$\|\eta\eta^\top\|_F^2 = Tr\left((\eta\eta^\top)^\top(\eta\eta^\top)\right) = Tr\left(\eta\eta^\top\eta\eta^\top\right) = \eta^\top\eta Tr(\eta\eta^\top) = (\eta^\top\eta)^2$$

Thus, we have $\|\eta\eta^\top\|_F^p = (\eta^\top\eta)^p = \|\eta\|_2^{2p}$. Since $\eta \sim \mathcal{N}(0, I_d)$ and thus, the vector $\eta$ has *i.i.d.* normal entries, thereby:

$$\|\eta\|_2^2 = \sum_i \eta_i^2 \sim \chi^2(d) \quad \implies \quad \mathbb{E}[\|\eta\|_2^{2p}] = \mathbb{E}[(X)^p] \quad \text{where } X \sim \chi^2(d)$$

where $\chi^2(d)$ denotes the chi-squared distribution with $d$ degrees of freedom. Now, we can just use the formulae for moments of $\chi^2(d)$, leading us to:

$$\mathbb{E}\left[\|\eta\|^{2p}\right] = \mathbb{E}[X^p] = 2^p \cdot \frac{\Gamma(p + \frac{d}{2})}{\Gamma(\frac{d}{2})} \overset{(a)}{\leq} 2^p\left(\frac{d}{2} + p\right)^p = (d + 2p)^p$$

where $\Gamma$ denotes the gamma function and for inequality $(a)$, we have just used the gamma function bound. $\qquad\square$

**Lemma A.8.** *We have:*

$$\mathbb{E}_{q_t}\left[\|s_r(t, z(t))\|^2\right] \leq \frac{d}{e^{2t} - 1}; \quad \mathbb{E}_{q_t}\left[\|s_r(t, z(t))\|^p\right] \leq \frac{(d + p)^{p/2}}{(e^{2t} - 1)^{p/2}}$$

$$\mathbb{E}_{q_t}\left[\|\nabla s_r(t, z)\|_F^2\right] \leq \frac{2d^2 + 6d}{(e^{2t} - 1)^2}$$

*Proof.* Similar to the Chen et al. (2023a), here we also utilize the fact that $\frac{y - z(t)}{\sqrt{1 - e^{-2t}}}$ is Gaussian. Using Lemma A.5 we have:

$$\mathbb{E}_{q_t}\left[\|s_r(t, z)\|^2\right] = \frac{1}{(e^{2t} - 1)}\mathbb{E}_{z \sim q_t}\left[\left\|\mathbb{E}_{y|z}\left[\frac{y - z}{\sqrt{e^{2t} - 1}}\right]\right\|^2\right] \leq \frac{1}{(e^{2t} - 1)}\mathbb{E}_{q_t}\mathbb{E}_{y|z}\left[\left\|\left[\frac{y - z}{\sqrt{e^{2t} - 1}}\right]\right\|^2\right]$$

$$= \frac{1}{(e^{2t} - 1)}\mathbb{E}_{\eta \sim \mathcal{N}(0, I_d)}\left[\|\eta\|^2\right]$$

$$= \frac{d}{e^{2t} - 1}$$

For a general $p \geq 2$, it becomes:

$$\mathbb{E}_{q_t}\left[\|s_r(t, z(t))\|^p\right] = \frac{1}{(e^{2t} - 1)^{p/2}}\mathbb{E}_{q_t}\left[\left\|\mathbb{E}_{y|z(t)}\left[\frac{y - z}{\sqrt{(e^{2t} - 1)}}\right]\right\|^p\right]$$

$$\leq \frac{1}{(e^{2t} - 1)^{p/2}}\mathbb{E}_{q_t}\mathbb{E}_{y|z(t)}\left[\left\|\left[\frac{y - z}{\sqrt{e^{2t} - 1}}\right]\right\|^p\right]$$

$$= \frac{1}{(e^{2t} - 1)^{p/2}}\mathbb{E}_{\eta \sim \mathcal{N}(0, I_d)}\left[\|\eta\|^p\right]$$

$$\leq \frac{(d + p)^{p/2}}{(e^{2t} - 1)^{p/2}} \qquad\qquad \text{(Lemma A.7)}$$

Going similarly, a *naive* bound on $\mathbb{E}_{q_t}\left[\|\nabla s_r(t,z)\|_F^2\right]$ will be:

$$\mathbb{E}_{z\sim q_t}\left[\|\nabla s_r(t,z)\|_F^2\right] = \mathbb{E}_{z\sim q_t}\left[\left\|\mathrm{Var}_{y|z}\left[\frac{y-z}{e^{2t}-1}\right] - \frac{I_d}{e^{2t}-1}\right\|_F^2\right] \qquad \text{(Lemma A.6)}$$

$$\leq 2\mathbb{E}_{z\sim q_t}\left[\left\|\mathbb{E}_{y|z}\left[\frac{y-z}{e^{2t}-1}\right]\left[\frac{y-z}{e^{2t}-1}\right]^\top\right\|_F^2\right] + \frac{2d}{(e^{2t}-1)^2}$$

$$\leq 2\mathbb{E}_{z\sim p_t}\mathbb{E}_{y|z}\left[\left\|\left[\frac{y-z}{e^{2t}-1}\right]\left[\frac{y-z}{e^{2t}-1}\right]^\top\right\|_F^2\right] + \frac{2d}{(e^{2t}-1)^2}$$

$$= 2\frac{1}{(e^{2t}-1)^2}\mathbb{E}_{\eta\sim\mathcal{N}(0,I_d)}\left[\|\eta\|^4\right] + \frac{2d}{(e^{2t}-1)^2}$$

$$= \frac{2d^2+6d}{(e^{2t}-1)^2}$$

$\square$

### A.3.1 EXPRESSING THE $\mathbb{E}_{p_t}\left[\|s'(t,x(t))\|^2\right]$ IN TERMS OF SPATIAL DERIVATIVES

Since we also need to bound $\partial_t s_r(t,z)$ to bound the $s'_r(t,z)$, we will utilise the Fokker Plank equation associated with the forward/reverse processes which relates the partial derivative w.r.t. t with the spatial derivative. The Fokker-Plank equation corresponding to the rescaled process $z(t)$ (Eq.13) would be:

$$\partial_t q_t(z) = -\sum_{i=1}^d \partial_i\left(-e^{2t}\nabla q_t(z)\right) = e^{2t}\Delta q_t(z) \qquad (17)$$

Since score function is just $\frac{\nabla q_t(z(t))}{q_t}$, we provide the corresponding *score-fpe* for the rescaled process to relate the $\partial_t s_r(t,z)$ with spatial derivative in the lemma below:

**Lemma A.9.** *We have the following counterpart of the Fokker-Planck equation for the score function of the rescaled process defined in Eq. 13:*

$$\partial_t s_r(t,z) = e^{2t}\Delta s_r(t,z) + e^{2t}\nabla\|s_r(t,z)\|^2 = e^{2t}\Delta s_r(t,z) + 2e^{2t}\nabla s_r(t,z)^\top s_r(t,z) \qquad (18)$$

*Proof.* To arrive at the equation for the score function, we first derive an equation for $\partial_t \log q_t$ by considering the following term:

$$e^{2t}\sum_{i=1}^d \partial_i\left(\nabla\log q_t(z)\right) = e^{2t}\sum_{i=1}^d \partial_i\left(\frac{\nabla q_t(z)}{q_t(z)}\right) = e^{2t}\sum_{i=1}^d\left(\frac{q_t(z)\partial_i\nabla q_t(z) - \partial_i q_t(z)\nabla q_t(z)}{p_t^2(z)}\right)$$

$$= e^{2t}\sum_{i=1}^d\left(\frac{\partial_i\nabla q_t(z)}{q_t(z)}\right) - e^{2t}\|\nabla\log q_t(z)\|^2$$

which results in:

$$\partial_t\log q_t(z) = \frac{\partial_t q_t(z)}{q_t(z)} = e^{2t}\sum_{i=1}^d\left(\frac{\partial_i\nabla q_t(z)}{q_t(z)}\right) = e^{2t}\sum_{i=1}^d \partial_i\left(\nabla\log q_t(z)\right) + e^{2t}\|\nabla\log q_t(z)\|^2$$

Now, again taking a spatial gradient:

$$\nabla\partial_t\log q_t(z) = e^{2t}\sum_{i=1}^d \nabla\partial_i\left(\nabla\log q_t(z)\right) + e^{2t}\nabla\|\nabla\log q_t(z)\|^2$$

Interchanging the operators result in the score Fokker-Planck equation for the forward process (on the reverse it would be negative):

$$\partial_t s_r(t, z) = e^{2t} \Delta s_r(t, z) + e^{2t} \nabla \|s_r(t, z)\|^2 = e^{2t} \Delta s_r(t, z) + 2e^{2t} \nabla s_r(t, z)^\top s_r(t, z)$$

□

Now based on this lemma, we express the overall derivative in terms of spatial derivative in the following lemma.

**Lemma A.10.** *We have the following relation for the overall score derivative $s_r'(t, z)$ and $\nabla s_r(t, z)$ for the rescaled process following the reverse ODE (Eq. 14):*

$$\mathbb{E}_{q_t}[\|s_r'(t, z)\|^2] = \mathbb{E}_{q_t}\left[e^{4t}\|\Delta s_r(t, z)\|_2^2 + e^{4t}\|\nabla s_r(t, z)^\top s_r(t, z)\|_2^2\right]$$
$$+ \mathbb{E}_{q_t}\left[2e^{4t}(\Delta s_r(t, z))^\top \left(\nabla s_r(t, z)^\top s_r(t, z)\right)\right] \quad (19)$$

*where (recall from Section 2.1) $\Delta$ denotes the Laplacian of a vector.*

*Proof.* Now, utilising the Fokker-Planck equation (FPE) for the score function, we will have:

$$\mathbb{E}_{q_t}[\|s_r'(t, z(t))\|^2]$$
$$= \mathbb{E}_{q_t}[s_r'(t, z)^\top s_r'(t, z)]$$
$$= \mathbb{E}_{q_t}\left[\left(\partial_t s_r(t, z) + \nabla s_r(t, z)^\top \left(\frac{dz}{dt}\right)\right)^\top \left(\partial_t s_r(t, z) + \nabla s_r(t, z)^\top \left(\frac{dz}{dt}\right)\right)\right]$$
$$\overset{(a)}{=} \mathbb{E}_{q_t}\left[\left(\partial_t s_r(t, z) - e^{2t} \nabla s_r(t, z)^\top s_r(t, z)\right)^\top \left(\partial_t s_r(t, z) - e^{2t} \nabla s_r(t, z)^\top s_r(t, z)\right)\right]$$
$$= \mathbb{E}_{q_t}\left[\|\partial_t s_r(t, z)\|_2^2 + e^{4t}\|\nabla s_r(t, z)^\top s_r(t, z)\|_2^2 - 2e^{2t}\partial_t s_r(t, z)^\top \left(\nabla s_r(t, z)^\top s_r(t, z)\right)\right]$$
$$\overset{(b)}{=} \mathbb{E}_{q_t}\left[\|e^{2t}\Delta s_r(t, z) + 2e^{2t}s_r(t, z)^\top \nabla s_r(t, z)\|_2^2 + e^{4t}\|\nabla s_r(t, z)^\top s_r(t, z)\|_2^2\right.$$
$$\left. - 2e^{2t}(e^{2t}\Delta s_r(t, z) + 2e^{2t}s_r(t, z)^\top \nabla s_r(t, z))^\top \left(\nabla s_r(t, z)^\top s_r(t, z)\right)\right]$$
$$= \mathbb{E}_{q_t}\left[e^{4t}\|\Delta s_r(t, z)\|_2^2 + e^{4t}\|\nabla s_r(t, z)^\top s_r(t, z)\|_2^2 + 2e^{4t}(\Delta s_r(t, z))^\top \left(\nabla s_r(t, z)^\top s_r(t, z)\right)\right]$$

where in $(a)$ we have used reverse ODE Eq. 14 and in $(b)$ we have used the FPE for score from Lemma A.9. □

### A.4 BOUNDING THE REQUIRED SPATIAL DERIVATIVE TERMS

Now to bound the terms $\mathbb{E}_{q_t}\left[\|\Delta s_r(t, z)\|_2^2\right]$, $\mathbb{E}\left[\|s_r(t, z)\nabla s_r(t, z)\|_2^2\right]$ appearing in Eq. 19, we analyze the relationship of these spatial gradient terms with $\frac{d}{dt}\mathbb{E}_{q_t}\left[\|s_r(t, z)\|^2\right]$. Since the naive bound on $\mathbb{E}_{q_t}\left[\|s_r(t, z)\|^2\right]$ is proportional to $d$ as against $d^2$ in case of $\mathbb{E}_{q_t}\left[\|\nabla s_r(t, z)\|_F^2\right]$, this can lead to improved $d-$dependence of the discretization error upon integrating this for a given time interval. We first discuss two lemmas: the first one establishes relation between $\frac{d}{dt}\mathbb{E}_{q_t}[\|s_r(t, z)\|^2]$ and $\mathbb{E}_{q_t}[\|\nabla s_r(t, z)\|_F^2]$. Then extending this lemma for general power $m$ in $\frac{d}{dt}\mathbb{E}_{q_t}[\|s_r(t, z)\|^m]$ leads to the terms comprising both $\nabla s_r$ and $s_r$ from which we can bound the term $\mathbb{E}\left[\|s_r(t, z)\nabla s_r(t, z)\|_2^2\right]$. Then, utilising these lemmas and bounding terms comprising $\delta q_t$ and $\nabla s_r$ in terms of $\frac{d}{dt}\mathbb{E}_{q_t}\left[\|s_r(t, z)\|^2\right]$, we finally bound $\mathbb{E}_{q_t}\left[\|\Delta s_r(t, z)\|_2^2\right]$ by applying Integration By Parts.

A.4.1 ESTABLISHING RELATION BETWEEN SCORE AND ITS FIRST ORDER SPATIAL GRADIENT TERMS

We first analyze the term $\frac{d}{dt}\mathbb{E}_{q_t}[\|s_r(t,z)\|^2]$ and manipulate it to relate it with $\mathbb{E}_{q_t}[\|\nabla s_r(t,z)\|_F^2]$ leading to the following lemma.

**Lemma A.11.** *We have:*

$$\frac{d}{dt}\mathbb{E}_{q_t}[\|s_r(t,z)\|^2] = -2e^{2t}\mathbb{E}_{q_t}[\|\nabla s_r(t,z)\|_F^2]$$

*Proof.* We begin by analysing the LHS term, taking the derivative inside the integral and utilise Fokker-Planck equation (FPE) (for $q_t$ and $s_r$ (Eq. 17, Eq. 18) for the rescaled process to convert it to spatial derivative and finally utilise Integration By Parts (IBP):

$$\frac{d}{dt}\mathbb{E}_{q_t}[\|s_r(t,z)\|^2]$$

$$= \frac{d}{dt}\int q_t(z)\|s_r(t,z)\|^2 dz$$

$$= \int \partial_t q_t(z)\|s_r(t,z)\|^2 dz + 2\int q_t(z)s_r(t,z)^\top \partial_t s_r(t,z)dz$$

$$\stackrel{(a)}{=} \int e^{2t}\Delta q_t(z)\|s_r(t,z)\|^2 dz + 2\int q_t(z)s_r(t,z)^\top \left(e^{2t}\Delta s_r(t,z) + e^{2t}\nabla\|s_r(t,z)\|^2\right)dz$$

$$\stackrel{(b)}{=} -e^{2t}\int \nabla q_t(z)\cdot\nabla\|s_r(t,z)\|^2 dz + 2e^{2t}\int q_t(z)s_r(t,z)^\top\left(\Delta s_r(t,z) + \nabla\|s_r(t,z)\|^2\right)dz$$

$$\stackrel{(c)}{=} e^{2t}\int \nabla q_t(z)\cdot\nabla\|s_r(t,z)\|^2 dz + 2e^{2t}\int q_t(z)s_r(t,z)^\top\left(\Delta s_r(t,z)\right)dz$$

$$= e^{2t}\int \nabla q_t(z)\cdot\nabla\|s_r(t,z)\|^2 dz + 2e^{2t}\int \underbrace{q_t(z)\Delta s_r(t,z)^\top}_{\text{jointly for IBP}} s_r(t,z)dz$$

$$\stackrel{(d)}{=} e^{2t}\int \nabla q_t(z)\cdot\nabla\|s_r(t,z)\|^2 dz - 2e^{2t}\int \nabla q_t(z)\cdot\nabla s_r(t,z)^\top s_r(t,z)dz$$

$$\qquad - 2e^{2t}\int q_t(z)\|\nabla s_r(t,z)\|_F^2 dz$$

$$\stackrel{(e)}{=} -2e^{2t}\int q_t(z)\|\nabla s_r(t,z)\|_F^2 dz$$

where in $(a)$ we have used FPE Eq. 17, 18, in $(b)$ we just use IBP, in $(c)$ we use $q_t(z)s_r(t,z) = \nabla q_t(z)$, in $(d)$ we again use IBP and in $(e)$ we use $\nabla\|s_r(t,z)\|^2 = 2(\nabla s_r(t,z))^\top s_r(t,z)$ so the first and second terms cancel out. □

We now generalize this lemma by establishing relation between $\frac{d}{dt}\mathbb{E}_{q_t}[\|s_r(t,z)\|^m]$ for a general $m \geq 2$ and $\mathbb{E}_{q_t}[\|\nabla s_r(t,z)\|_F^2]$. For $m > 2$, the RHS should have terms comprising both $\nabla s_r(t,z), s_r(t,z)$ and thus the result can be used to bound the second part of the RHS in Eq. 19.

**Lemma A.12.** *We have the following general result for the score function $s_r(t, z_t)$ of the rescaled process $z_t$ defined in Eq. 13, holding for any $m > 2$:*

$$e^{-2t}\frac{d}{dt}\mathbb{E}_{q_t}[\|s_r(t,z)\|_2^m] = -m\mathbb{E}_{q_t}[\|s_r(t,z)\|_2^{m-2}\|\nabla s_r(t,z)\|_F^2]$$

$$- \frac{m(m-2)}{4}\mathbb{E}_{q_t}\left[\|s_r(t,z)\|_2^{m-4}\left\|\left(\nabla\|s_r(t,z)\|_2^2\right)\right\|_2^2\right]$$

*Proof.* Here also, we start with analyzing the LHS similar to previous lemma (as discussed in Section 2.1, $\partial_i$ corresponds to partial derivative w.r.t. $i^{th}$ coordinate of $z$, $\Delta = \sum_i \partial_i\partial_i$ is the Laplacian,

$s_r(\cdot)_i$ corresponds to $i^{th}$ element of $s_r$ which implies $\|s_r(t,z)\|^2 = \sum_{i=1}^d s_r^2(t,z)_i)$ (all the variables under $\sum$ range from $1$ to $d$ if not mentioned):

$$e^{-2t}\frac{d}{dt}\mathbb{E}_{q_t}[\|s_r(t,z)\|_2^m]$$

$$\stackrel{(a)}{=} \int e^{-2t}\partial_t q_t(z)\|s_r(t,z)\|_2^m dz + m\int e^{-2t}q_t(z)\|s_r(t,z)\|_2^{m-2}s_r(t,z)^\top\partial_t s_r(t,z)dz$$

$$\stackrel{(b)}{=} \sum_{i=1}^d \int \partial_i\partial_i q_t(z)\|s_r(t,z)\|_2^m dz$$

$$+ m\int q_t(z)\|s_r(t,z)\|_2^{m-2}\left(\sum_{i,j=1}^d s_r(t,z)_j\left(\partial_i\partial_i s_r(t,z)_j + \partial_i s_r^2(t,z)_j\right)\right)dz$$

$$\stackrel{(c)}{=} -\frac{m}{2}\sum_{i,j=1}^d \int \partial_i q_t(z)\|s_r(t,z)\|_2^{m-2}\partial_i s_r^2(t,z)_j dz$$

$$+ m\sum_{i,j=1}^d \int q_t(z)\|s_r(t,z)\|_2^{m-2}s_r(t,z)_j\left(\partial_i\partial_i s_r(t,z)_j + \partial_i s_r^2(t,z)_j\right)dz$$

$$\stackrel{(d)}{=} m\sum_{i,j}\int q_t(z)\|s_r(t,z)\|^{m-2}s_r(t,z)_i s_r(t,z)_i\partial_i s_r(t,z)_j dz$$

$$+ m\sum_{i,j}\int \underbrace{q_t(z)\|s_r(t,z)\|_2^{m-2}s_r(t,z)_j}_{I_1}\partial_i\partial_i s_r(t,z)_j dz$$

$$\stackrel{(e)}{=} m\sum_{i,j}\int q_t(z)\|s_r(t,z)\|^{m-2}s_r(t,z)_i s_r(t,z)_i\partial_i s_r(t,z)_j dz$$

$$- m\sum_{i,j}\int \partial_i q_t(z)\cdot\|s_r(t,z)\|_2^{m-2}s_r(t,z)_j\partial_i s_r(t,z)_j dz$$

$$- m\sum_{i,j}\int q_t(z)\cdot\partial_i\|s_r(t,z)\|_2^{m-2}\cdot s_r(t,z)_j\partial_i s_r(t,z)_j dz$$

$$- m\sum_{i,j}\int p_t(z)\cdot\|s_r(t,z)\|_2^{m-2}\cdot\partial_i s_r(t,z)_j\partial_i s_r(t,z)_j dz$$

$$\stackrel{(f)}{=} -m(m-2)\sum_{i,j,k}\int p_t(z)\cdot\|s_r(t,z)\|_2^{m-4}\cdot s_r(t,z)_k\partial_i s_r(t,z)_k s_r(t,z)_j\cdot\partial_i s_r(t,z)_j dz$$

$$- m\sum_{i,j}\int q_t(z)\cdot\|s_r(t,z)\|_2^{m-2}\cdot\left(\partial_i s_r(t,z)_j\right)^2 dz$$

$$= -\frac{m(m-2)}{4}\mathbb{E}_{q_t}\left[\|s_r(t,z)\|_2^{m-4}\left\|\left(\nabla\|s_r(t,z)\|_2^2\right)\right\|_2^2\right] - m\mathbb{E}_{q_t}\left[\|s_r(t,z)\|_2^{m-2}\|\nabla s_r(t,z)\|_F^2\right]$$

where in $(a)$ we have used $\partial_t\|s_r(t,z)\|_2^m = m\|s_r(t,z)\|_2^{m-2}s_r(t,z)^\top\partial_t s_r(t,z)$, $(b)$ implies the use of FPEs Eq. 17, Lemma A.9 , $(c)$ is the application of Integration By Parts on the first term, $(d)$ uses $\partial_i q_t(z) = q_t(z)s_r(t,z)_i$ then subtract it from the second part of the second term and use $\partial_i s_r^2(t,z)_j = 2s_r(t,z)_j\partial_i s_r(t,z)_j$, $(e)$ implies again using Integration By Parts on the second term where one term is jointly considered as $I_1$ and the other remaining. $(f)$ is derived using $\partial_t q = q_t(z)s_r(t,z)$ on second term, cancelling the first two terms and writing $\partial_i\|s_r(t,z)\|_2^{m-2} = (m-2)\|s_r(t,z)\|_2^{m-4}\sum_{k=1}^d s_r(t,z)_k\partial_i s_r(t,z)_k$. $\qquad\square$

Now, as discussed before, a consequence lemma of this lemma is that we can bound the second term comprising both $s_r$ and $\nabla s_r$ in our main Eq. 19 (since the RHS contains these terms for a general m). This is stated as a lemma below.

**Lemma A.13.** *Defining* $X_m = \int q_t(z)\|s_r(t,z)\|^{m-2}\|\nabla s_r(t,z)\|_F^2 dz$, $X_m' = \int q_t(z)\|s_r(t,z)\|^{m-4}\|\nabla\|s_r(t,z)\|_2^2\|_2^2 dz$ *we can bound it as follows for any* $m > 2$*:*

$$X_m \leq -\frac{1}{m}e^{-2t}\frac{d}{dt}\mathbb{E}_{q_t}[\|s_r(t,z)\|^m], \quad X_m' \leq -\frac{4}{m(m-2)}e^{-2t}\frac{d}{dt}\mathbb{E}_{q_t}[\|s_r(t,z)\|_2^m]$$

*Proof.* For this, considering the Lemma A.12, we have:

$$\frac{-1}{m}e^{-2t}\frac{d}{dt}\mathbb{E}_{q_t}[\|s_r(t,z)\|_2^m] = \underbrace{\mathbb{E}_{q_t}[\|s_r(t,z)\|_2^{m-2}\|\nabla s_r(t,z)\|_F^2]}_{X_m}$$

$$+ \frac{(m-2)}{4}\underbrace{\mathbb{E}_{q_t}\left[\|s_r(t,z)\|_2^{m-4}\left\|\left(\nabla\|s_r(t,z)\|_2^2\right)\right\|_2^2\right]}_{X_m'}$$

When $m > 2$, we will have $X_m, X_m' \geq 0$, thus, $X_m \leq -\frac{1}{m}e^{-2t}\frac{d}{dt}\mathbb{E}_{q_t}[\|s_r(t,z)\|^m]$ and $X_m' \leq -\frac{4}{m(m-2)}e^{-2t}\frac{d}{dt}\mathbb{E}_{q_t}[\|s_r(t,z)\|^m]$. $\square$

### A.4.2 BOUNDING THE SECOND ORDER SPATIAL GRADIENT OF SCORE TERM

For this, we first provide two lemmas to bound terms comprising second order spatial derivative of the law $q_t$ and first order spatial derivative of the score $s_r$.

**Lemma A.14.** *For the rescaled process $z(t)$ in Eq. 13, we have:*

$$\frac{\Delta q_t(z)}{q_t(z)} = \frac{-d}{(e^{2t}-1)} + \mathbb{E}_{y|z}\left[\frac{\|y-z\|_2^2}{(e^{2t}-1)^2}\right]$$

*Proof.* Here also similar to the score function calculation in Lemma A.5, we have:

$$\frac{\Delta q_t(z)}{q_t(z)} = \frac{\nabla\cdot\nabla q_t(z)}{q_t(z)} \overset{(a)}{=} \frac{\nabla\cdot\int\frac{y-z}{(e^{2t}-1)}e^{-\frac{\|z-y\|^2}{2(e^{2t}-1)}}p_{data}(y)dy}{\int e^{-\frac{\|z-y\|^2}{2(e^{2t}-1)}}p_{data}(y)dy}$$

$$= \frac{\int\left(\nabla\cdot\frac{y-z}{(e^{2t}-1)}\right)e^{-\frac{\|z-y\|^2}{2(e^{2t}-1)}}p_{data}(y)dy + \int\left(\nabla e^{-\frac{\|z-y\|^2}{2(e^{2t}-1)}}p_{data}(y)\right)\frac{y-z}{(e^{2t}-1)}dy}{\int e^{-\frac{\|z-y\|^2}{2(e^{2t}-1)}}p_{data}(y)dy}$$

$$= \frac{\int\frac{-d}{(e^{2t}-1)}e^{-\frac{\|z-y\|^2}{2(e^{2t}-1)}}p_{data}(y)dy + \int e^{-\frac{\|z-y\|^2}{2(e^{2t}-1)}}p_{data}(y)\left(\frac{y-z}{(e^{2t}-1)}\right)^\top\frac{y-z}{(e^{2t}-1)}dy}{\int e^{-\frac{\|z-y\|^2}{2(e^{2t}-1)}}p_{data}(y)dy}$$

$$= \frac{-d}{(e^{2t}-1)} + \int P(y|z)\left(\frac{y-z}{(e^{2t}-1)}\right)^\top\frac{y-z}{(e^{2t}-1)}dy$$

$$= \frac{-d}{(e^{2t}-1)} + \mathbb{E}_{y|z}\left[\frac{\|y-z\|_2^2}{(e^{2t}-1)^2}\right]$$

where in $(a)$ we have taken the expression also used in Lemma A.5 and as discussed before $y \sim p_{data}$. $\square$

**Lemma A.15.** *For the rescaled process $z(t)$, defined in Eq. 13, we have the following bound for the term involving the Laplacian of margin and Jacobian of score:*

$$\int \Delta q_t(z)\|\nabla s_r(t,z)\|_F^2 dz \leq \frac{C_d d^2}{(e^{2t}-1)^3} - \frac{e^{-2t}de}{2(1+\frac{1}{\log d})(e^{2t}-1)}\frac{d}{dt}\mathbb{E}_{q_t}[\|s_r(t,z)\|^2]$$

*where $C_d = \frac{(1+2\frac{\log d}{d}+\frac{6}{d})^{\log d+3}}{(1+\log d)}$.*

*Proof.* We start by writing Laplacian as $\sum_i \partial_i\partial_i$ and decomposing the term as follows:

$$\int \Delta q_t(z)\|\nabla s_r(t,z)\|_F^2 dz = \int \sum_i \partial_i\partial_i q_t(z)\|\nabla s_r(t,z)\|_F^2 dz$$

$$= \int \sum_i \partial_i\partial_i q_t(z)\|\nabla s_r(t,z)\|_F^2 dz$$

$$= \int q_t(z)J_t(z)\|\nabla s_r(t,z)\|_F^2 dz - \frac{d}{e^{2t}-1}\mathbb{E}_{q_t}\left[\|\nabla s_r(t,z)\|_F^2\right]$$

$$= \int q_t(z)J_t(z)c_m^{-1}\|\nabla s_r(t,z)\|_F^{2/l}\cdot c_m\|\nabla s_r(t,z)\|_F^{2/m}dz$$

$$- \frac{d}{e^{2t}-1}\mathbb{E}_{q_t}\left[\|\nabla s_r(t,z)\|_F^2\right]$$

$$\overset{(b)}{\leq} \frac{c_m^{-l}}{l}\int q_t(z)J_t^l(z)\|\nabla s_r(t,z)\|_F^2 dz + \frac{c_m^m}{m}\int q_t(z)\|\nabla s_r(t,z)\|_F^2 dz$$

$$- \frac{d}{e^{2t}-1}\mathbb{E}_{q_t}\left[\|\nabla s_r(t,z)\|_F^2\right]$$

where $J_t(z) = \frac{\sum_i \partial_i\partial_i q_t(z)}{q_t(z)} + \frac{d}{e^{2t}-1}$, $l$ and $m$ are constants where $l>1$, $1/l+1/m=1$ and in $(b)$, we have just used $ab \leq \frac{1}{l}a^l + \frac{1}{m}b^m$ with $a = J_t(z)c_m^{-1}$, $b = c_m$. Now, we utilise Lemma A.6 for the spatial gradient of score, Lemma A.14 to write $J_t(z) = -\frac{d}{e^{2t}-1} + \mathbb{E}_{y|z}\left[\frac{\|y-z\|_2^2}{(e^{2t}-1)^2}\right]$. Also, for the second term, we can just use Lemma A.11, leading to :

$$= \frac{c_m^{-l}}{l}\int q_t(z)\left(\mathbb{E}_{y|z}\left[\frac{\|y-z\|_2^2}{(e^{2t}-1)^2}\right]\right)^l\left\|\mathrm{Var}_{y|z}\left[\frac{y-z}{e^{2t}-1}\right] - \frac{I_d}{e^{2t}-1}\right\|_F^2 dz$$

$$- \frac{e^{-2t}}{2}\left(\frac{c_m^m}{m} - \frac{d}{e^{2t}-1}\right)\frac{d}{dt}\mathbb{E}_{q_t}[\|s_r(t,z)\|^2]$$

$$\leq \frac{c_m^{-l}}{l}\mathbb{E}_{q_t}\left[\left(\mathbb{E}_{y|z}\left[\frac{\|y-z\|_2^2}{(e^{2t}-1)^2}\right]\right)^l\left(\left(\mathbb{E}_{y|z}\left[\frac{\|y-z\|_2^2}{(e^{2t}-1)^2}\right]\right)^2 + \frac{d}{(e^{2t}-1)^2}\right)\right]$$

$$- \frac{e^{-2t}}{2}\left(\frac{c_m^m}{m} - \frac{d}{e^{2t}-1}\right)\frac{d}{dt}\mathbb{E}_{q_t}[\|s_r(t,z)\|^2]$$

$$\leq \frac{c_m^{-l}}{l}\mathbb{E}_{q_t}\mathbb{E}_{y|z}\left[\left(\left[\frac{\|y-z\|_2^2}{(e^{2t}-1)^2}\right]\right)^{l+2}\right] + \frac{d}{(e^{2t}-1)^2}\frac{c_m^{-l}}{l}\mathbb{E}_{q_t}\mathbb{E}_{y|z}\left[\left(\left[\frac{\|y-z\|_2^2}{(e^{2t}-1)^2}\right]\right)^l\right]$$

$$- \frac{e^{-2t}}{2}\left(\frac{c_m^m}{m} - \frac{d}{e^{2t}-1}\right)\frac{d}{dt}\mathbb{E}_{q_t}[\|s_r(t,z)\|^2] \qquad \text{(Jensen's Inequality)}$$

where in the last step we first multiplied the terms in the first part and then used Jensen's inequality for each. Now, utilising the fact that $\frac{y-z}{\sqrt{e^{2t}-1}} \sim \mathcal{N}(0, I_d)$ and using the Gaussian moment bound

from Lemma A.7, the last term can be further rewritten and bounded as:

$$
= \frac{c_m^{-l}}{l(e^{2t}-1)^{l+2}} \mathbb{E}_{\eta\sim\mathcal{N}(0,I_d)}\left[\|\eta\|_2^{2l+4} + d\|\eta\|_2^{2l}\right] - \frac{e^{-2t}}{2}\left(\frac{c_m^m}{m} - \frac{d}{e^{2t}-1}\right)\frac{d}{dt}\mathbb{E}_{q_t}[\|s_r(t,z)\|^2]
$$

$$
\leq \frac{c_m^{-l}}{l(e^{2t}-1)^{l+2}} \cdot (d+2l+4)^{l+2} - \frac{e^{-2t}}{2}\left(\frac{c_m^m}{m} - \frac{d}{e^{2t}-1}\right)\frac{d}{dt}\mathbb{E}_{q_t}[\|s_r(t,z)\|^2]
$$

$$
= \frac{\left(\frac{d}{(e^{2t}-1)^{1/m}}\right)^{-l}}{l(e^{2t}-1)^{l+2}} \cdot (d+2l+4)^{l+2} - \frac{e^{-2t}}{2}\left(\frac{\left(\frac{d}{(e^{2t}-1)^{1/m}}\right)^m}{m} - \frac{d}{e^{2t}-1}\right)\frac{d}{dt}\mathbb{E}_{q_t}[\|s_r(t,z)\|^2]
$$

$$
(c_m = \tfrac{d}{(e^{2t}-1)^{1/m}})
$$

$$
= \frac{d^2 \cdot (1+2l/d+4/d)^{l+2}}{l(e^{2t}-1)^{l+2-l/m}} - \frac{e^{-2t}}{2}\left(\frac{d^m}{m(e^{2t}-1)} - \frac{d}{e^{2t}-1}\right)\frac{d}{dt}\mathbb{E}_{q_t}[\|s_r(t,z)\|^2]
$$

$$
= \frac{d^2}{(e^{2t}-1)^3} \cdot \underbrace{\frac{(1+2\frac{\log d}{d}+\frac{6}{d})^{\log d+3}}{(1+\log d)}}_{C_d} - \frac{e^{-2t}}{2}\left(\frac{d^{1+\frac{1}{\log d}}}{(1+\frac{1}{\log d})(e^{2t}-1)} - \frac{d}{e^{2t}-1}\right)\frac{d}{dt}\mathbb{E}_{q_t}[\|s_r(t,z)\|^2]
$$

$$
(l = 1+\log d, \, m = 1 + \tfrac{1}{\log d})
$$

$$
\leq \frac{C_d d^2}{(e^{2t}-1)^3} - \frac{e^{-2t}de}{2(1+\frac{1}{\log d})(e^{2t}-1)}\frac{d}{dt}\mathbb{E}_{q_t}[\|s_r(t,z)\|^2]
$$

$$
(d^{\frac{1}{\log d}} = e)
$$

where we have used $c_m = \frac{d}{(e^{2t}-1)^{1/m}}$, $l = 1+\log d$, $m = 1 + \frac{1}{\log d}$ which results in $C_d = \frac{(1+2\frac{\log d}{d}+\frac{6}{d})^{\log d+3}}{(1+\log d)}$. $\qquad\square$

### A.4.3 BOUND THE LAPLACIAN OF THE SCORE

Now, using the previous two lemmas and the Lemma A.13, we bound the second order spatial derivative term of the score function in the following lemma.

**Lemma A.16.** *We have the following bound for the second order score derivative term in Eq. 19:*

$$
\mathbb{E}_{q_t}\left[\|\Delta s_r(t,z)\|_2^2\right] \leq \frac{32C_d d^2}{13(e^{2t}-1)^3} - \frac{de^{-2t}}{(e^{2t}-1)}\frac{d}{dt}\mathbb{E}_{q_t}[\|s_r(t,z)\|^2] - \frac{8}{13}e^{-2t}\frac{d}{dt}\mathbb{E}_{q_t}\left[\|\nabla s_r(t,z)\|_F^2\right]
$$

$$
- \frac{20e^{-2t}}{13}\frac{d}{dt}\mathbb{E}_{q_t}\left[\|s_r(t,z)\|_2^4\right]
$$

*where $C_d$ is defined in Lemma A.15.*

*Proof.* The proof is just a careful utilization of the integration by parts, Fokker-Planck equation (FPE) and the reverse ODE Eq. 14. The proof starts with manipulating the term: $\frac{d}{dt}\mathbb{E}_{q_t}\left[\|\nabla s_r(t,z)\|_F^2\right]$ to break it down in the target term and remaining terms from the previous two lemmas and Lemma A.13. Then the target term term is expressed via this term and the remaining terms where we replace the bounds for the remaining terms from the mentioned lemmas. It is

as follows (again we use $\partial_i$ for the derivative w.r.t. $i^{th}$ coordinate and the Laplacian by $\sum_i \partial_i \partial_i$):

$$\frac{d}{dt} \int q_t(z) \|\nabla s_r(t,z)\|_F^2 dz$$

$$= \int \partial_t q_t(z) \|\nabla s_r(t,z)\|_F^2 dz + \int q_t(z) \partial_t \sum_{i,j} \partial_j s_r^2(t,z)_i dz$$

$$= \int e^{2t} \sum_{i=1}^d \partial_i \partial_i q_t(z) \|\nabla s_r(t,z)\|_F^2 dz + \int q_t(z) \left( 2 \sum_{i,j=1}^d \partial_j s_r(t,z)_i \partial_j \partial_t s_r(t,z)_i \right) dz$$

$$\text{(Eq. 17 for first term)}$$

$$\overset{(a)}{=} \int e^{2t} \sum_i \partial_i \partial_i q_t(z) \|\nabla s_r(t,z)\|_F^2 dz$$

$$+ 2e^{2t} \int q_t(z) \underbrace{\sum_{i,j} \partial_j s_r(t,z)_i \partial_j \left( \sum_k \partial_k \partial_k s_r(t,z)_i + \sum_k \partial_i s_r^2(t,z)_k \right)}_{I_1} dz$$

$$\overset{(b)}{=} \int e^{2t} \sum_i \partial_i \partial_i q_t(z) \|\nabla s_r(t,z)\|_F^2 dz - 2e^{2t} \sum_{i,j,k} \int \partial_j q_t(z) \partial_j s_r(t,z)_i \left( \partial_k \partial_k s_r(t,z)_i + \partial_i s_r^2(t,z)_k \right) dz$$

$$- 2e^{2t} \sum_{i,j,k} \int q_t(z) \partial_j \partial_j s_r(t,z)_i \left( \partial_k \partial_k s_r(t,z)_i + \partial_i s_r^2(t,z)_k \right) dz$$

$$\overset{(c)}{=} e^{2t} \left( \int \sum_i \partial_i \partial_i q_t(z) \|\nabla s_r(t,z)\|_F^2 dz \right.$$

$$- 2 \sum_i \int q_t(z) \sum_j s_r(t,z)_j \partial_i s_r(t,z)_j \sum_k \left( \partial_k \partial_k s_r(t,z)_i + \partial_i s_r^2(t,z)_k \right) dz$$

$$\left. - 2 \sum_i \int q_t(z) \sum_j \partial_j \partial_j s_r(t,z)_i \sum_k \left( \partial_k \partial_k s_r(t,z)_i + \partial_i s_r^2(t,z)_k \right) dz \right)$$

where $(a)$ implies use of Lemma A.9 for the second term, $(b)$ implies using Integration By Parts for the second term where we consider the term $I_1$ as one part and the remaining as other, $(c)$ implies using $\partial_j q_t(z) = q_t(z) s_r(t,z)_j$ and then $\partial_j s_r(t,z)_i = \partial_i s_r(t,z)_j$ in the second term. Now, we consider the terms except first, treating $\sum_j \partial_i s_r^2(t,z)_j = 2 \sum_j s_r(t,z)_j \partial_i s_r(t,z)_j = b_i$ and $\sum_j \partial_j \partial_j s_r(t,z)_i = a_i$ these terms can be written as:

$$= \sum_i -b_i \cdot (a_i + b_i) - 2a_i(a_i + b_i) = \sum_i -2a_i^2 - b_i^2 - 3a_ib_i \leq \sum_i -2a_i^2 - b_i^2 + \frac{3}{8}a_i^2 + 6b_i^2$$

which leads us to:

$$e^{-2t} \frac{d}{dt} \mathbb{E}_{q_t} \left[ \|\nabla s_r(t,z)\|_F^2 \right] \leq \underbrace{\int \sum_i \partial_i \partial_i q_t(z) \|\nabla s_r(t,z)\|_F^2 dz}_{T_0} - \frac{13}{8} \underbrace{\sum_i \int q_t(z) \left( \sum_j \partial_j \partial_j s_r(t,z)_i \right)^2 dz}_{T_1 \text{ (target term)}}$$

$$+ 20 \underbrace{\sum_i \int q_t(z) \left( \sum_j s_r(t,z)_j \partial_j s_r(t,z)_i \right)^2 dz}_{T_2}$$

Denoting the first term in the RHS as $T_0$, second or the target term as $T_1$ and third term as $T_2$, we have the following expression for our target term $T_1$ (rewriting $\sum_i \partial_i \partial_i$ as Laplacian operator):

$$\frac{13}{8} \mathbb{E}_{q_t} \left[ \|\Delta s_r(t,z)\|_2^2 \right]$$

$$\leq T_0 + 20 T_2 - e^{-2t} \frac{d}{dt} \mathbb{E}_{q_t} \left[ \|\nabla s_r(t,z)\|_F^2 \right]$$

$$= T_0 + 5 \mathbb{E}_{q_t} \left[ \|\nabla\|s_r(t,z)\|_2^2\|_2^2 \right] - e^{-2t} \frac{d}{dt} \mathbb{E}_{q_t} \left[ \|\nabla s_r(t,z)\|_F^2 \right] \qquad \text{(rewriting } T_2\text{)}$$

$$\overset{(a)}{\leq} \frac{4d^2}{(e^{2t}-1)^3} - \frac{e^{-2t} de}{2(1+\frac{1}{\log d})(e^{2t}-1)} \frac{d}{dt} \mathbb{E}_{q_t}[\|s_r(t,z)\|^2] - \frac{5}{2} e^{-2t} \frac{d}{dt} \mathbb{E}_{q_t} \left[ \|s_r(t,z)\|_2^4 \right]$$

$$\qquad\qquad - e^{-2t} \frac{d}{dt} \mathbb{E}_{q_t} \left[ \|\nabla s_r(t,z)\|_F^2 \right]$$

where in step $(a)$ we have just used Lemma A.15 for $T_0$ term and the observation that the third term (obtained by rewriting $T_2$) is just the $X'_q$ in Lemma A.13 for $q = 4$ which we bound using the Lemma A.13. Now since $d > 1$, we have approximated the value $\frac{4e}{13(1+\frac{1}{\log d})} < 1$ (since $\frac{d}{dt} \mathbb{E}_{q_t}[\|s_r(t,z)\|^2]$ is negative, can be seen from Lemma A.13) leading to the final bound. □

### A.4.4 BOUNDING THE DISCRETIZATION ERROR FOR EACH INTERVAL

Utilising the lemmas discussed above for bounding the spatial derivaitve terms in Eq. 19, we now provide a lemma which using these bounds provides a final aggregated bound for $\mathbb{E}_{q_t}[\|s'_r(t,z)\|_2^2]$.

**Lemma A.17.** *For the rescaled function, we have the following bound for* $\mathbb{E}_{q_t}[\|s'_r(t,z)\|_2^2]$*:*

$$\mathbb{E}_{q_t}[\|s'_r(t,z)\|_2^2] \leq \frac{40 C_d d^2 e^{4t}}{13(e^{2t}-1)^3} - \frac{5 e^{2t} de}{4(e^{2t}-1)} \frac{d}{dt} \mathbb{E}_{q_t}[\|s_r(t,z)\|^2] - \frac{10}{13} e^{2t} \frac{d}{dt} \mathbb{E}_{q_t} \left[ \|\nabla s_r(t,z)\|_F^2 \right]$$

$$- 3 e^{2t} \frac{d}{dt} \mathbb{E}_{q_t}[\|s_r(t,z)\|^4]$$

*where $C_d$ is taken from Lemma A.15.*

*Proof.*

$$\mathbb{E}_{q_t}[\|s'_r(t,z)\|^2]$$

$$= \mathbb{E}_{q_t} \left[ e^{4t} \|\Delta s_r(t,z)\|_2^2 + e^{4t} \|\nabla s_r(t,z)^\top s_r(t,z)\|_2^2 + 2 e^{4t} (\Delta s_r(t,z))^\top \left( \nabla s_r(t,z)^\top s_r(t,z) \right) \right]$$

$$\leq \mathbb{E}_{q_t} \left[ \frac{5}{4} e^{4t} \|\Delta s_r(t,z)\|_2^2 + 5 e^{4t} \|\nabla s_r(t,z)^\top s_r(t,z)\|_2^2 \right] \quad (2a \cdot b \leq \frac{\|a\|^2}{4} + 4\|b\|^2 \text{ for } a,b \in \mathbb{R}^d)$$

$$= \frac{5}{4} e^{4t} \mathbb{E}_{q_t} \left[ \|\Delta s_r(t,z)\|_2^2 \right] + \frac{5}{4} e^{4t} \mathbb{E}_{q_t} \left[ \|\nabla\|s_r(t,z)\|_2^2\|_2^2 \right]$$

$$\leq \frac{5}{4} e^{4t} \mathbb{E}_{q_t} \left[ \|\Delta s_r(t,z)\|_2^2 \right] - \frac{5}{8} e^{2t} \frac{d}{dt} \mathbb{E}_{q_t}[\|s_r(t,z)\|^4]$$

$$\leq \frac{40 C_d d^2 e^{4t}}{13(e^{2t}-1)^3} - \frac{5 e^{2t} d}{4(e^{2t}-1)} \frac{d}{dt} \mathbb{E}_{q_t}[\|s_r(t,z)\|^2] - \frac{10}{13} e^{2t} \frac{d}{dt} \mathbb{E}_{q_t} \left[ \|\nabla s_r(t,z)\|_F^2 \right]$$

$$- \frac{265}{104} e^{2t} \frac{d}{dt} \mathbb{E}_{q_t}[\|s_r(t,z)\|^4]$$

where the first equality uses Lemma A.10, the second last inequality uses $X'_m$ $(m = 4)$ bound from Lemma A.13 and the last inequality uses Lemma A.16 for the first term. Now since $\frac{d}{dt} \mathbb{E}_{q_t}[\|s_r(t,z)\|^4]$ will be negative from Lemma A.13, then here we can use $\frac{265}{104} < 3$ leading to the final bound. □

Now, we have the following Lemma for bounding the discretization error $z(t)$: $\mathbb{E} \left[ \|z_{k-0.5} - \tilde{z}_{k-0.5}\|_2^2 \right]$.

**Lemma A.18.** *The discretization error for each interval* $\mathbb{E}\left[\|z_{k-0.5} - \tilde{z}_{k-0.5}\|_2^2\right]$ *discussed Lemma A.3 can be bounded as (where $h'_k = h_k + h_{k-1}$ and recall $t_{k-2} = t_k - h_k - h_{k-1}$):*

$$e^{-2t_{k-2}}\mathbb{E}\left[\|z_{k-0.5} - \tilde{z}_{k-0.5}\|_2^2\right] \leq \frac{\left((2C_d + 10)d^2 + 24d\right)(h'_k)^3 e^{h'_k}(e^{h'_k} - 1)}{(1 - e^{-2t_{k-2}})^3}$$

$$- \frac{(h'_k)^3}{2}e^{h'_k}\left[e^{4t}\left(\frac{10}{13}\mathbb{E}_{q_t}\left[\|\nabla s_r(t, z)\|_F^2\right] + 3\mathbb{E}_{q_t}[\|s_r(t, z)\|^4]\right)\right]_{t_{k-2}}^{t_k}$$

$$- \frac{5(h'_k)^3 e^{h'_k} d}{8(1 - e^{-2t_{k-2}})}\left[e^{2t}\mathbb{E}_{q_t}[\|s_r(t, z)\|^2]\right]_{t_{k-2}}^{t_k}$$

*Proof.* Using Lemma A.4 and Lemma A.17, it can be bounded as

$$e^{-2t_{k-2}}\mathbb{E}\left[\|z_{k-0.5} - \tilde{z}_{k-0.5}\|_2^2\right]$$

$$\leq e^{-2t_{k-2}}\frac{1}{2}(h_k + h_{k-1})^3 \int_{t_{k-2}}^{t_k} e^{4t}\mathbb{E}\left[\|s'_r(t, z(t))\|_2^2\right]dt$$

$$\leq \frac{(h_k + h_{k-1})^3}{2}e^{h_k + h_{k-1}} \int_{t_{k-2}}^{t_k} e^{2t}\mathbb{E}_{q_t}\left[\|s_r(t, z(t))\|_2^2\right]dt$$

$$\leq \frac{(h_k + h_{k-1})^3}{2}e^{h_k + h_{k-1}} \int_{t_{k-2}}^{t_k}\left(\frac{40C_d d^2 e^{6t}}{13(e^{2t} - 1)^3} - \frac{5e^{4t}d}{4(e^{2t} - 1)}\frac{d}{dt}\mathbb{E}_{q_t}[\|s_r(t, z)\|^2]\right)dt$$

$$- \frac{(h_k + h_{k-1})^3}{2}e^{h_k + h_{k-1}} \int_{t_{k-2}}^{t_k}\left(e^{4t}\left(\frac{10}{13}\frac{d}{dt}\mathbb{E}_{q_t}\left[\|\nabla s_r(t, z)\|_F^2\right] + \frac{d}{dt}\mathbb{E}_{q_t}[\|s_r(t, z)\|^4]\right)\right)dt$$

$$\overset{(c)}{\leq} \frac{20C_d d^2(h_k + h_{k-1})^3 e^{h_k + h_{k-1}}(e^{h_k + h_{k-1}} - 1)}{13(1 - e^{-2t_{k-2}})^3}$$

$$- \frac{5(h_k + h_{k-1})^3 e^{h_k + h_{k-1}} d}{8(1 - e^{-2t_{k-2}})}\left(\left[e^{2t}\mathbb{E}_{q_t}[\|s_r(t, z)\|^2]\right]_{t_{k-2}}^{t_k} - \int_{t_{k-2}}^{t_k} 2e^{2t}\mathbb{E}_{q_t}[\|s_r(t, z)\|^2]dt\right)$$

$$- \frac{(h_k + h_{k-1})^3}{2}e^{h_k + h_{k-1}}\left[e^{4t}\left(\frac{10}{13}\mathbb{E}_{q_t}\left[\|\nabla s_r(t, z)\|_F^2\right] + 3\mathbb{E}_{q_t}[\|s_r(t, z)\|^4]\right)\right]_{t_{k-2}}^{t_k}$$

$$+ (h_k + h_{k-1})^3 e^{h_k + h_{k-1}} \int_{t_{k-2}}^{t_k} 2e^{4t}\left(\frac{10}{13}\mathbb{E}_{q_t}\left[\|\nabla s_r(t, z)\|_F^2\right] + 3\mathbb{E}_{q_t}[\|s_r(t, z)\|^4]\right)dt$$

$$\overset{(d)}{\leq} \frac{20C_d d^2(h_k + h_{k-1})^3(e^{h_k + h_{k-1}} - 1)}{13(1 - e^{-2t_{k-2}})^3} + \frac{10(h_k + h_{k-1})^3 e^{h_k + h_{k-1}} d}{8(1 - e^{-2t_{k-2}})}\int_{t_{k-2}}^{t_k} e^{2t}\frac{d}{e^{2t} - 1}dt$$

$$+ (h_k + h_{k-1})^3 e^{h_k + h_{k-1}} \int_{t_{k-2}}^{t_k} e^{4t}\left(\frac{20(2d^2 + 6d)}{13(e^{2t} - 1)^2} + \frac{6d^2 + 12d}{(e^{2t} - 1)^2}\right)dt$$

$$- \frac{(h_k + h_{k-1})^3}{2}e^{h_k + h_{k-1}}\left(\left[e^{4t}\left(\frac{10}{13}\mathbb{E}_{q_t}\left[\|\nabla s_r(t, z)\|_F^2\right] + 3\mathbb{E}_{q_t}[\|s_r(t, z)\|^4]\right)\right]_{t_{k-2}}^{t_k}\right)$$

$$- \frac{5(h_k + h_{k-1})^3 e^{h_k + h_{k-1}} d}{8(1 - e^{-2t_{k-2}})}\left[e^{2t}\mathbb{E}_{q_t}[\|s_r(t, z)\|^2]\right]_{t_{k-2}}^{t_k}$$

$$\leq \frac{\left((2C_d + 10)d^2 + 24d\right)(h_k + h_{k-1})^3 e^{h_k + h_{k-1}}(e^{h_k + h_{k-1}} - 1)}{(1 - e^{-2t_{k-2}})^3}$$

$$- \frac{(h_k + h_{k-1})^3}{2}e^{h_k + h_{k-1}}\left[e^{4t}\left(\frac{10}{13}\mathbb{E}_{q_t}\left[\|\nabla s_r(t, z)\|_F^2\right] + 3\mathbb{E}_{q_t}[\|s_r(t, z)\|^4]\right)\right]_{t_{k-2}}^{t_k}$$

$$- \frac{5(h_k + h_{k-1})^3 e^{h_k + h_{k-1}} d}{8(1 - e^{-2t_{k-2}})}\left[e^{2t}\mathbb{E}_{q_t}[\|s_r(t, z)\|^2]\right]_{t_{k-2}}^{t_k}$$

where $(c)$ uses $\int \frac{e^{6t}}{(e^{2t} - 1)^3}dt \leq \frac{e^{4t_{k-2}}}{e^{4t_{k-2}} - 1}\int_{t_{k-2}}^{t_k}\frac{e^{2t}}{e^{2t} - 1}dt$, $\int_{t_{k-2}}^{t_k}\frac{e^{2t}}{e^{2t} - 1}dt = \frac{1}{2}\log\left(\frac{e^{2t_k} - 1}{e^{2t_{k-2}} - 1}\right)$, $\log(1 + x) \leq x$ for the first term and implies applying the Integration By Parts to the second and third terms,

where in the second term, we have considered $e^{2t}\frac{d}{dt}\mathbb{E}_{q_t}[\|s_r(t,z)\|^2]$ as one term and use the max value of the remaining term since we know from Lemma A.11 that $e^{2t}\frac{d}{dt}\mathbb{E}_{q_t}[\|s_r(t,z)\|^2] \le 0$, in step $(d)$ we recollect the integral terms and since they have a positive contribution, just replace the term inside the integral with the upper bound from Lemma A.8. In the last step, similar to step $(c)$, we have used $\int_{t_{k-2}}^{t_k} \frac{e^{2t}}{e^{2t}-1}dt = \frac{1}{2}\log\left(\frac{e^{2t_k}-1}{e^{2t_{k-2}}-1}\right)$, for $\int_{t_{k-2}}^{t_k}\frac{e^{4t}}{(e^{2t}-1)^2}dt \le \frac{e^{2t_{k-2}}}{e^{2t_{k-2}}-1}\int_{t_{k-2}}^{t_k}\frac{e^{2t}}{e^{2t}-1}dt$ and finally $\log(1+x) \le x$ .

$\square$

## A.5 Proving Theorem 3.1

We first discuss a lemma based on standard calculus which would be utilized in the theorem proof.

**Lemma A.19.** *For some fixed $c \in (0,\frac{1}{2})$ and denoting $a := 2(1-c)$, $b := 2-c$. For $x \in (0,1)$ define*

$$f(x) := \frac{(2-c)^3\,x^3\,e^{bx}}{e^{ax}-1}, \qquad g(x) := \frac{(2-c)^3\,x^3\,e^{bx}}{(e^{ax}-1)\left(1-e^{-\frac{a^2}{2c}x}\right)}.$$

*Then $f$ and $g$ are increasing on $(0,1)$ and there exist absolute constants $C_f, C_g > 0$ (independent of $c$ and $x$) such that for all $x \in (0,1)$,*

$$0 \le f(x) - f\left((1-c)^2x\right) \le C_f\,c\,x^2, \qquad 0 \le g(x) - g\left((1-c)^2x\right) \le C_g\,\frac{c}{1-e^{-\frac{2(1-c)^2}{c}x}}\,x^2.$$

*Proof.* Using $\frac{d}{dx}\log(e^{\alpha x}-1) = \frac{\alpha e^{\alpha x}}{e^{\alpha x}-1}$,

$$(\log f)'(x) = \frac{3}{x}+b-\frac{ae^{ax}}{e^{ax}-1}, \qquad (\log g)'(x) = \frac{3}{x}+b-\frac{ae^{ax}}{e^{ax}-1}-\frac{\frac{a^2}{2c}}{e^{\frac{a^2}{2c}x}-1}.$$

For $t > 0$ we have the elementary bound $\frac{1}{e^t-1} \le \frac{1}{t}$; hence

$$\frac{ae^{ax}}{e^{ax}-1} \le a+\frac{1}{x} \qquad \frac{\frac{a^2}{2c}}{e^{\frac{a^2}{2c}x}-1} \le \frac{1}{x}.$$

Therefore, for $x \in (0,1)$,

$$(\log f)'(x) \ge \frac{3}{x}+b-\left(a+\frac{1}{x}\right) = \frac{2}{x}+c \ge \frac{2}{x} > 0,$$

and

$$(\log g)'(x) \ge \frac{3}{x}+b-\left(a+\frac{1}{x}\right)-\frac{1}{x} = \frac{1}{x}+c \ge \frac{1}{x} > 0.$$

Hence $f$ and $g$ are increasing on $(0,1)$. Using $e^{ax}-1 \ge ax$ and $(2-c)^3 \le 8$, $b \le 2$, we get for $x \in (0,1)$

$$f(x) = \frac{(2-c)^3 x^3 e^{bx}}{e^{ax}-1} \le \frac{8\,x^3\,e^2}{ax} \le 8e^2\,x^2.$$

Consequently,

$$g(x) = \frac{f(x)}{1-e^{-\frac{a^2}{2c}x}} \le \frac{8e^2\,x^2}{1-e^{-\frac{a^2}{2c}x}}$$

From above, we have:

$$f'(x) = f(x)\,(\log f)'(x) \le f(x)\left(\frac{3}{x}+b\right) \le 8e^2x^2\left(\frac{3}{x}+2\right) \le 40e^2x.$$

Similarly,

$$g'(x) = g(x)\,(\log g)'(x) \le g(x)\left(\frac{3}{x}+b\right) \le 8e^2\,\frac{x^2}{1-e^{-\frac{a^2}{2c}x}}\left(\frac{3}{x}+2\right) \le 40e^2\,\frac{x}{1-e^{-\frac{a^2}{2c}x}},$$

Now for $y := (1-c)^2 x \in (0, x)$, $x - y = \left(1 - (1-c)^2\right)x = (2c - c^2)x \leq 2cx$. By the mean value theorem, for some $\xi \in (y, x) \subset (0, 1)$,

$$f(x) - f(y) = f'(\xi)\,(x - y) \leq 40e^2\,\xi\,(2cx) \leq 80e^2\,c\,x^2,$$

which yields $f(x) - f((1-c)^2 x) \leq C_f\,c\,x^2$ where $C_f = 80e^2$ is an absolute constant. Likewise, for some $\eta \in (y, x)$,

$$g(x) - g(y) = g'(\eta)\,(x - y) \leq 40e^2\,\frac{\eta}{1 - e^{-\frac{a^2}{2c}\eta}}\,(2cx) \leq 80e^2\,\frac{c}{1 - e^{-\frac{a^2}{2c}x}}\,x^2,$$

since $t \mapsto 1 - e^{-t}$ is increasing and $\eta \leq x$. This gives

$$g(x) - g\left((1-c)^2 x\right) \leq C_g\,\frac{c}{1 - e^{-\frac{2(1-c)^2}{c}x}}\,x^2$$

where $C_g = 80e^2$ is an absolute constant. $\qquad\square$

**Proof of Theorem 3.1.** Now, using the Lemmas discussed above, we provide the proof for Theorem 3.1.

*Proof.* We first bound the discretization error term for the rescaled process in Lemma A.18 aggregated across all the intervals. For this, using $h'_k = h_k + h_{k-1}$, we bound it as following:

$$\sum_{k=2}^{K+1} \frac{e^{-2t_{k-2}}}{e^{2h_{k-1}} - 1}\mathbb{E}\left[\|z_{k-0.5} - \tilde{z}_{k-0.5}\|_2^2\right]$$

$$\leq \sum_{k=2}^{K+1} \frac{\left((2C_d + 10)d^2 + 24d\right)(h'_k)^3 e^{h'_k}(e^{h'_k} - 1)}{(e^{2h_{k-1}} - 1)(1 - e^{-2t_{k-2}})^3}$$

$$+ \sum_{k=2}^{K+1} \frac{(h'_k)^3 e^{h'_k}}{2(e^{2h_{k-1}} - 1)}\left[e^{4t}\left(\frac{10}{13}\mathbb{E}_{q_t}\left[\|\nabla s_r(t, z)\|_F^2\right] + 3\mathbb{E}_{q_t}[\|s_r(t, z)\|^4]\right)\right]_{t_k}^{t_{k-2}}$$

$$+ \sum_{k=2}^{K+1} \frac{(h'_k)^3 e^{h'_k}}{2(e^{2h_{k-1}} - 1)}\left[\frac{5de^{2t}}{4(1 - e^{-2t_{k-2}})}\mathbb{E}_{q_t}\left[\|s_r(t, z)\|^2\right]\right]_{t_k}^{t_{k-2}}$$

$$= \sum_{k=2}^{K+1} \frac{\left((2C_d + 10)d^2 + 24d\right)(h'_k)^3 e^{h'_k}(e^{h'_k} - 1)}{(e^{2h_{k-1}} - 1)(1 - e^{-2t_{k-2}})^3} + \frac{(h'_2)^3 e^{h'_2}}{e^{2h_1} - 1}R(t_0)$$

$$+ \frac{(h'_3)^3 e^{h'_3}}{e^{2h_2} - 1}R(t_1) + \frac{(h'_2)^3 e^{h'_2}}{(e^{2h_1} - 1)(1 - e^{-2t_0})}R_1(t_0) + \frac{(h'_3)^3 e^{h'_3}}{(e^{2h_2} - 1)(1 - e^{-2t_1})}R_1(t_1)$$

$$+ \sum_{k=2}^{K-1}\left(\frac{(h'_{k+2})^3 e^{h'_{k+2}}}{e^{2h_{k+1}} - 1} - \frac{(h'_k)^3 e^{h'_k}}{(e^{2h_{k-1}} - 1)}\right)R(t_k) - \frac{(h'_{K+1})^3}{(e^{2h_K} - 1)}R(t_{K+1}) - \frac{(h'_K)^3}{e^{2h_{K-1}} - 1}R(t_K)$$

$$+ \sum_{k=2}^{K-1}\left(\frac{(h'_{k+2})^3 e^{h'_{k+2}}}{(e^{2h_{k+1}} - 1)(1 - e^{-2t_k})} - \frac{(h'_k)^3 e^{h'_k}}{(e^{2h_{k-1}} - 1)(1 - e^{-2t_{k-2}})}\right)R_1(t_k)$$

$$- \frac{(h'_{K+1})^3 e^{h'_{K+1}}}{(e^{2h_K} - 1)(1 - e^{-2t_{K-1}})}R_1(t_{K+1}) - \frac{(h'_K)^3 e^{h'_K}}{(e^{2h_{K-1}} - 1)(1 - e^{-2t_{K-2}})}R_1(t_K)$$

where $R(t) = \frac{1}{2}e^{4t}\left(\frac{10}{13}\mathbb{E}_{q_t}\left[\|\nabla s_r(t, z)\|_F^2\right] + 3\mathbb{E}_{q_t}[\|s_r(t, z)\|^4]\right) \geq 0$, $R_1(t) = \frac{5de^{2t}}{8}\mathbb{E}_{q_t}[\|s_r(t, z)\|^2] \geq 0$.

**Selecting the step size.** Now for the mentioned choice of the step size $h_k = t_k - t_{k-1} = c\min\{1, t_k\}$, we will have $t_{k-1} = (1-c)t_k$, $h_{k-1} = (1-c)h_k$ when $t_k \leq 1$ and $h_k = c$ for remaining. Since $t_0 = \delta$, we will have:

$$\delta = (1-c)^M; \quad T - 1 = c(K + 1 - M)$$

for some $M \leq K + 2$ with $t_M = 1$. Thus, we will have $c \lesssim \frac{\log(\frac{1}{\delta}) + T}{K}$ and will have a very small value for the mentioned condition $K \geq d(\frac{1}{\delta} + T)$. Also, for the coefficients of terms containing $R$, for $t_k \leq 1$ we will have $t_{k-1} = (1-c)t_k, h_{k-1} = (1-c)h_k, h_{k-2} = (1-c)^2 h_k$ and thus, we will have:

$$\frac{(h_{k+2} + h_{k+1})^3 e^{h_{k+2} + h_{k+1}}}{e^{2h_{k+1}} - 1} - \frac{(h_k + h_{k-1})^3 e^{h_k + h_{k-1}}}{e^{2h_{k-1}} - 1} = \left(\frac{(2-c)^3 h_{k+2}^3 e^{(2-c)h_{k+2}}}{e^{2(1-c)h_{k+2}} - 1} - \frac{(2-c)^3 h_k^3 e^{(2-c)h_k}}{e^{2(1-c)h_k} - 1}\right) \text{ for}$$

$k$ when $t_{k+2} \leq 1$ and $0$ for the rest. This can be written as $f(h_{k+2}) - f(h_k)$ where $f(x) = \frac{(2-c)^3 x^3 e^{(2-c)x}}{e^{(1-c)x} - 1}$ would be an increasing function w.r.t. $x$ for $x < 1$ in the small $c$ region ($c < 0.5$). For this, we will also have $f(x) - f((1-c)^2 x) \lesssim cx^2$ (Lemma A.19). Similarly for the $R_1$, we have to consider: $g(x) = \frac{(2-c)^3 x^3 e^{(2-c)x}}{(e^{2(1-c)x} - 1)\left(1 - e^{-\frac{2(1-c)^2 x}{c}}\right)}$ and it will also be increasing on $(0, 1)$ for small $c$ ($c < 0.5$) and $g(x) - g((1-c)^2 x) \lesssim \frac{c}{1 - e^{-\frac{2(1-c)^2 x}{c}}} x^2$ from Lemma A.19. Since $h_k(\text{¿}0)$ is an increasing sequence, we can use the upper bound for $R(t), R_1(t)$ using Lemma A.8 as:

$$R(t) \leq \frac{4d^2 + 11d}{(1 - e^{-2t})^2}; \qquad R_1(t) \leq \frac{5d^2}{8(1 - e^{-2t})}$$

Also, the term $C_d$ from Lemma A.15 is $C_d = \frac{(1 + 2\frac{\log d}{d} + \frac{6}{d})^{\log d + 3}}{(1 + \log d)} \leq 12$ for $d \geq 10$ and thus we will have $C_d$ as $O(1)$. Since $R(t), R_1(t) \geq 0$, the negative terms corresponding to $R(t_K), R(t_{K+1}), R1(t_K), R1(t_{K+1})$ can be dropped and we will finally have:

$$\sum_{k=2}^{K+1} \frac{e^{-2t_{k-2}}}{e^{2h_{k-1}} - 1} \mathbb{E}\left[\|z_{k-0.5} - \tilde{z}_{k-0.5}\|_2^2\right]$$

$$\leq \sum_{k=2}^{K+1} \frac{\left((2C_d + 10)d^2 + 24d\right)(h_k')^3 e^{h_k'}(e^{h_k'} - 1)}{(e^{2h_{k-1}} - 1)(1 - e^{-2t_{k-2}})^3}$$

$$+ \frac{(h_2')^3 e^{h_2'}}{e^{2h_1} - 1} R(t_0) + \frac{(h_3')^3 e^{h_3'}}{e^{2h_2} - 1} R(t_1) + \frac{(h_2')^3 e^{h_2'}}{(e^{2h_1} - 1)(1 - e^{-2t_0})} R_1(t_0)$$

$$+ \frac{(h_3')^3 e^{h_3'}}{(e^{2h_2} - 1)(1 - e^{-2t_1})} R_1(t_1) + \sum_{k=2}^{K-1}\left(\frac{(h_{k+2}')^3 e^{h_{k+2}'}}{e^{2h_{k+1}} - 1} - \frac{(h_k')^3 e^{h_k'}}{(e^{2h_{k-1}} - 1)}\right) R(t_k)$$

$$+ \sum_{k=2}^{K-1}\left(\frac{(h_{k+2}')^3 e^{h_{k+2}'}}{(e^{2h_{k+1}} - 1)(1 - e^{-2t_k})} - \frac{(h_k')^3 e^{h_k'}}{(e^{2h_{k-1}} - 1)(1 - e^{-2t_{k-2}})}\right) R_1(t_k)$$

$$\lesssim \sum_{k=2}^{K+1} \frac{d^2 h_k^3}{(1 - e^{-2t_{k-2}})^3} + h_2^2\left(R(t_0) + \frac{R_1(t_0)}{1 - e^{-2t_0}}\right) + h_3^2\left(R(t_1) + \frac{R_1(t_1)}{1 - e^{-2t_1}}\right)$$

$$+ \sum_{k=2}^{M} ch_{k+2}^2\left(R(t_k) + \frac{R_1(t_k)}{1 - e^{-2t_k}}\right) + \sum_{k=M+1}^{K-1} \frac{c^3 R_1(t_k)}{(1 - e^{-2t_{k-2}})^2}$$

$$\lesssim \sum_{k=2}^{M} \frac{d^2 c^3 t_k^3}{(1 - e^{-2t_{k-2}})^3} + \sum_{k=M+1}^{K+1} \frac{d^2 c^3}{(1 - e^{-2t_{k-2}})^3} + \frac{c^2 t_2^2}{(1 - e^{-2t_0})^2} + \frac{c^2 t_3^2}{(1 - e^{-2t_1})^2}$$

$$\lesssim \sum_{k=2}^{K+1} d^2 c^3$$

where $t_k \leq 1$ for $k \leq M$. Now using Lemma A.2, Lemma A.1, Lemma A.3 and the scaling back the above bound on the aggregated error for the rescaled process $\tilde{z}$, we will have (using $u < e^u - 1 < 2u$

for $u \in (0,1)$):

$$\mathrm{KL}\left(p_{t_1}\middle\|\hat{p}_{t_1}\right)$$

$$\leq \mathrm{KL}\left(p_{t_{K+1}}\middle\|\hat{p}_{t_{K+1}}\right) + \mathbb{E}_{p_{t_1,\ldots,t_{K+1}}}\left[\sum_{k=2}^{K+1} \mathrm{KL}\left(p_{t_{k-1}|t_k}(\cdot|x_k)\middle\|\hat{p}_{t_{k-1}|t_k}(\cdot|x_k)\right)\right]$$

$$= \mathrm{KL}\left(p_{t_{K+1}}\middle\|\hat{p}_{t_{K+1}}\right) + \sum_{k=2}^{K+1} \frac{e^{-2h_{k-1}}}{1-e^{-2h_{k-1}}}\mathbb{E}\|x_{k-0.5} - \tilde{x}_{k-0.5}\|_2^2$$

$$+ \frac{e^{-2h_{k-1}}}{1-e^{-2h_{k-1}}}(e^{h_k+h_{k-1}} - 1)^2 \mathbb{E}[\|s(t_k, x_k) - \hat{s}(t_k, x_k)\|^2]$$

$$\lesssim \mathrm{KL}\left(p_{t_{K+1}}\middle\|\hat{p}_{t_{K+1}}\right) + \sum_{k=2}^{K+1} d^2 c^3 + \sum_{k=2}^{K+1} h_k \mathbb{E}[\|s(t_k, x_k) - \hat{s}(t_k, x_k)\|^2]$$

The last term can be just bounded using Assumption 2.1 and the first term is the initialization error discussed below.

**Initialization Error.** The term $\mathrm{KL}\left(p_{t_{K+1}}\middle\|\hat{p}_{t_{K+1}}\right)$ is the error due to initializing the generation using the Normal distribution and can be bounded via convergence of the forward OU process after the total time $T$ (Chen et al., 2023c):

$$\mathrm{KL}\left(p_{t_{K+1}}\middle\|\hat{p}_{t_{K+1}}\right) \leq (d + m_2)e^{-T}$$

where $m_2 = \mathbb{E}[\|x_0\|^2]$. Thus, we have the final expression as:

$$\mathrm{KL}\left(p_{t_1}\middle\|\hat{p}_{t_1}\right) \lesssim (d + m_2)e^{-T} + d^2 c^3 K + T\varepsilon_{score}^2$$

$\square$

