# OpenReview forum: "A Sharp KL Convergence Analysis for Diffusion Models under Minimal Assumptions"
_ICLR.cc/2026/Conference — ICLR 2026 Poster_

### Official Review · Reviewer_EBD6 · 2025-10-27

**Soundness:** 2
**Presentation:** 3
**Contribution:** 2
**Rating:** 6
**Confidence:** 4

**Summary:**

The paper theoretically proves that $O\left(\frac{d}{\varepsilon}\right)$ diffusion step is sufficient to reach KL divergence $O\left(\varepsilon^{2}\right)$ to a blurred target distribution by taking a probability-flow ODE step then adding a small forward-noising step. It improves on prior $O\left(\tfrac{d}{\varepsilon^{2}}\right)$ iteration complexity rates under similar assumptions on time averaged score-estimation error. This is based on the observation that the discretization error contribution to KL divergence drops like $O(\frac{1}{K^{2}})$ as the number of steps $K$ increases.

**Strengths:**

The paper presents strong theoretical analysis with genuinely new ideas. The framework and bounds are carefully developed: assumptions are explicit. It is a well written paper that improves on previously known iteration complexity under minimal score estimation assumption.

**Weaknesses:**

There is no experiment validating whether the improved bounds translate to any practical improvements.

**Questions:**

Experimentally showing that the discretization error contribution to KL divergence drops like $O(\frac{1}{K^{2}})$ would put the paper on much stronger footing.

---

> ### Author Response · Authors · 2025-11-18
>
> We thank the reviewer for their thoughtful feedback. We agree that providing experimental evidence for the discretization-error decay would further strengthen the paper. However, we would like to respectfully emphasize that the primary aim of this work is to offer a theoretical analysis that helps explain the empirical effectiveness of DDPM samplers.
>
> For real-world tasks such as image generation, obtaining the true score function is challenging, which makes it difficult to empirically measure the discretization error associated with the intermediate process described in Eqs. (9)–(10). However, if the reviewer has recommendations for a feasible experimental setup where the true score is accessible, we would be happy to consider conducting such an analysis.

---

> > ### Comment · Reviewer_EBD6 · 2025-11-25
> >
> > Thank you for the clarification. An Ornstein–Uhlenbeck forward process with a Gaussian mixture initial distribution would be an ideal setup for numerical studies: in this case the exact score is available in closed form, so one can run the sampler with the true score, eliminate the score-estimation error, and thereby isolate the $K$-scaling of the discretization error (up to a $K$-independent initialization term).

---

> > > ### Author Response · Authors · 2025-12-02
> > >
> > > Thanks for suggesting this experimental setup. We would like to point the reviewer the following work [1] has already provided a synthetic experimental setup for the gaussian distribution for the first and second order discretization of the DDPM sampler. From the figure 1 in the paper, it can be observed that for the standard DDPM sampler the KL divergence approximately varies as $O(\frac{\log^2 K}{K^2})$, which addresses the reviewers experimental verfication of the discretization error concern. We can add a reference to this if required.
> > >
> > > [1]. Gen Li, and Changxiao Cai. Provable acceleration for diffusion models under minimal assumptions. arXiv preprint arXiv:2410.23285, 2024.

---

### Official Review · Reviewer_LSV4 · 2025-10-31

**Soundness:** 2
**Presentation:** 2
**Contribution:** 3
**Rating:** 6
**Confidence:** 3

**Summary:**

The paper shows that a hybrid sampling scheme consisting of a two time step backward ODE followed by one time step of forward SDE leads to a KL error that can be bound proportional to $1/\varepsilon_{\text{score}}$, improving on Benton et al (2023/4) that establishes a bound in $1/\varepsilon^2_{\text{score}}$. This error bound only concerns the distribution at some stopping time $t_1$.

**Strengths:**

* The proposed sampling scheme is simple, with a single score approximation by iteration. However, it does not correspond to the dynamic of a continuous equation (see Question).
* The convergence result improves on Benton et al. regarding dependency on $\varepsilon_{\text{score}}$.

**Weaknesses:**

The paper presentation should be improved.

Algorithm 1 lacks of clarity:
* line 2: Why sample from $t_K$ not $T = t_{K+1}$ (line 236) as a start using the standard normal distribution (line 238)?
* For the last step $k=1$, what is used to define $h_0 = t_{0}-t_{-1}$?
* What is the purpose of lines 9 and 10 recalling notation of distributions within the algorithm?

Also Theorem 3.1 is badly stated regarding stepsize/time discretization. $t_0<t_1<...$ are first given as a fixed discretization sequence, but then $h_k$ is given thus fixing the $t_k$s by a fixed rule.

l. 180: Empirical counterpart: Should initialization be mentioned here? Also confusing notation $t_{k}$ and $p_{t-1|t}$ in the same line.


Minor remarks:
* l. 076: dependence or dependency?
* l. 111: DDPM "showed that the corresponding denoising kernels are also Gaussian" It is my understanding that the kernels are chosen to be Gaussian by design.
* l. 161: The dependcy in $t$ should also appear in $\varepsilon$, or speak only of the marginal of the process not the OU process itself.
* l. 180: donot
* l. 209: Say that $z(t)$ is the variance exploding counterpart
* l. 214: space issue
* l. 248: Algorithm For Diffusion -> for
* l. 316: +1 is missing in first term of RHS.
* l. 354: $T_{est}$ police issue for est
* Several references are ArXiv of published papers, eg Benton et al 2023 -> ICLR 2024, Song et al 2023 -> ICML 2024.

**Questions:**

The proposed scheme is:
Two time step backward EI ODE followed by one time step of forward SDE:

$$
\hat{x}'_{k-0.5} = \exp(h\_{k}+h\_{k-1}) \hat{x}'\_{k} + (\exp (h\_{k}+h\_{k-1})-1)s(t_k,\hat{x}'\_{k})
$$

$$
\hat{x}'\_{k-1} = \exp (-h\_{k-1}) \hat{x}'\_{k-0.5} + \sqrt{1-e^{-2 h\_{k-1}}}\eta_k.
$$
This can be written as a one step scheme:
$$
\hat{x}'\_{k-1} = \exp (h\_{k}) \hat{x}'\_{k} + (\exp (h\_{k})- \exp (-h\_{k-1}))s(t\_k,\hat{x}'\_{k}) + \sqrt{1-e^{-2 h\_{k-1}}}\eta_k.
$$

This looks close to a convex combination of a backward EI step for the SDE and a backward EI step for the ODE (although not exactly). My question is: Can you interpret the proposed scheme as the discretization of a backward SDE?

---

> ### Author Response · Authors · 2025-11-18
>
> We thank the reviewer for the review and a detailed feedback on the paper presentation. The issues with the clarity of Algorithm 1 have been resolved where the line 2 in the algorithm samples form t_{K+1} now. The lines 9 and 10 in the algorithm which were previously just given for convenience have been removed now to avoid confusion.
> We have also updated the statement of theorem 3.1 to make it more clear.
>
> Line 180: Yes we agree initialization should be mentioned here and have updated it. The confusing notation issue has been resolved.
>
> The minor remarks have been addressed and the citations have also been updated where we now use published versions for most and arxiv of some. The issue mention with line 111 of the original submission has also been addressed (line 118 of the revised).
>
> All these changes have been marked in red.
>
> ***Regarding interpreting the proposed scheme as a discretization of the backward SDE***: Thanks for the question. Since [1] showed that corresponding to the reverse SDE (Eq. 2 in our paper), there exists a probability flow ODE (Eq. 3) which has the same marginal distribution, so we can exactly simulate the reverse step of size h_k along the reverse SDE as a step with size $h_k + h_{k-1}$ along the true probability flow ODE and then travelling forward along the OU process for with size $h_{k-1}$. This decomposition leads to the interpretation of the true reverse SDE as a larger probability flow step followed by noise addition along the forward process.
>
> If we use the Exponential Integrator discretization scheme for going along the ODE with step size $h_k + h_{k-1}$ and then follow the exact dynamics for the forward (OU) process for step size $h_{k-1}$, this will lead us to the update scheme we have considered in the paper for the discretization error analysis (Eq. 9,10) and upon using the approximate score will lead to the update scheme in Eq.7,8 or Algorithm 1. So for this setup, the continuous time dynamics behind our update will be a larger step ($h_k+ h_{k-1}$) along the probability flow ODE and one step forward ($h_{k-1}$) which, when using the true score, will be exactly equal in the distribution as one step ($h_{k}$) along the reverse SDE and thus, is just an interpretation of the reverse SDE.
>
> Therefore, the continuous time dynamics (in absence of any score estimation error) behind our update scheme matches the distribution of the true reverse SDE for this diffusion setup. We would be happy to provide any further clarifications on this and if required can also write this formally in our paper.
>
> [1] Score-based generative modeling through stochastic differential equations. International Conference on Learning Representations, 2020.

---

> > ### Author Response · Authors · 2025-11-21
> >
> > We once again thank the reviewer for their detailed feedback and are following up to confirm whether the clarifications provided above resolve the concerns raised. We would be happy to offer further explanation or make additional updates to the draft if needed.

---

> > > ### Comment · Reviewer_LSV4 · 2025-11-25
> > >
> > > I acknowledge that I read the response of the authors.
> > >
> > > * The correction in Eq. (1) (line 172) is not satisfying in my opinion.  The OU process involves some stochastic integral to define the $\varepsilon(t)$ term.  See Eq (2.1) of [1] or Propositon 1 of [2] for the exact expression.
> > >
> > > * Quoting your answer:
> > > >> we can exactly simulate the reverse step of size $h_k$ along the reverse SDE as a step with size $h_{k}+h_{k-1}$
> > >  along the true probability flow ODE and then travelling forward along the OU process for with size
> > > $h_{k-1}$.
> > >
> > > I don't understand this point, it seems to me that there is a confusion regarding the term *exactly simulate*. Does it mean sampling exactly the reverse SDE dynamic OR sampling some process that share the same time marginal as the reverse SDE?
> > >
> > >
> > > References:
> > >
> > > [1] Wasserstein Convergence Guarantees for a General Class of Score-Based Generative Models
> > > Xuefeng Gao, Hoang M. Nguyen, Lingjiong Zhu; 26(43):1−54, JMLR, 2025.
> > >
> > > [2] Diffusion models for Gaussian distributions: Exact solutions and Wasserstein errors
> > > Emile Pierret, Bruno Galerne Proceedings of the 42nd International Conference on Machine Learning, PMLR 267:49355-49381, 2025.

---

> ### Author Response · Authors · 2025-11-26
>
> We once again thank the reviewer for raising their concerns. Below we address both of them
>
> 1.) Regarding the correction of Eq. (1): The eq. we have provided for our diffusion setup (line 171) corresponds to the case when $\beta(t)=2$ in the VP case in the reference [1] the reviewer has provided, which results in $f(t)=1, g(t)=\sqrt{2}$. Now, substituting these values in Eq. (2.1) of [1] and then solving the integral will lead to Eq. (1) of our draft. This SDE considered in line 171 is standard in literature and for reference the reviewer can also refer to the last paragraph of the first column in the second page of [2] which also considers this SDE and has the same forward process as ours.
>
> 2.) By the term exactly simulate in the discussion above, we mean exactly sampling some process that share the same time marginal as the reverse SDE.
>
> We would be happy to discuss further and provide any additional details if needed.
>
> [1]  Wasserstein Convergence Guarantees for a General Class of Score-Based Generative Models Xuefeng Gao, Hoang M. Nguyen, Lingjiong Zhu; 26(43):1−54, JMLR, 2025.
>
> [2] Improved analysis of score-based generative modeling: User-friendly bounds under minimal smoothness assumptions. In International Conference on Machine Learning, pp. 4735–4763. PMLR, 2023a.

---

### Official Review · Reviewer_mkej · 2025-11-04

**Soundness:** 2
**Presentation:** 2
**Contribution:** 1
**Rating:** 2
**Confidence:** 4

**Summary:**

This paper investigates the convergence rate of DDPM in terms of KL divergence and establishes an iteration complexity of $O(d/\varepsilon)$, improving upon previous results in Benton et al. (2023). In particular, the authors first quantify the discretization error via the corresponding ODE and then translate this error into a KL divergence bound by injecting noise into the current sample.

**Strengths:**

The theoretical result provides an improved convergence rate in KL divergence compared with Benton et al. (2023).

**Weaknesses:**

The main contribution of this paper lies in extending the previously established total variation bound to a KL divergence bound. However, this extension appears to be of limited significance, as it does not improve the dependence on either the dimensionality or the target accuracy.

More importantly, the authors claim to introduce a novel technique that **controls the KL divergence for the probability flow ODE with injected noise through the squared $l_2$ discretization error of the probability flow ODE (as stated in Lemma A.1)**. Nevertheless, this observation has been widely recognized and utilized in prior works, which the authors fail to acknowledge.
A lot of researches has focused on reducing the KL divergence by designing ODE-based sampling methods aimed at improving the squared $l_2$ discretization error of the probability flow ODE. For example, [1] investigated a second-order strategy, [2,3] analyzed the randomized midpoint method, and they improve the convergence rate of diffusion models. While it is fair to note that this paper is among the first to apply such a technique to the first-order method (i.e., DDPM), the claimed methodological novelty is overstated.

In summary, given the limited theoretical advancement and the overstatement of technical novelty, I find it difficult to provide a positive assessment.

[1]. Gen Li, and Changxiao Cai. Provable acceleration for diffusion models under minimal assumptions. arXiv preprint arXiv:2410.23285, 2024.

[2]. Shivam Gupta, Linda Cai, and Sitan Chen. Faster Diffusion Sampling with Randomized Midpoints: Sequential and Parallel. arXiv preprintarXiv:2406.00924, 2024.

[3] Gen Li, and Yuchen Jiao. Improved convergence rate for diffusion probabilistic models. The Thirteenth International Conference on Learning Representations, 2024.

**Questions:**

Please refer to the Weakness part.

---

> ### Author Response · Authors · 2025-11-18
>
> We respectfully but firmly disagree with the claim that the work lacks novelty. The prior reference [1] you mentioned controls the error in Wasserstein distance using an ODE sampler and then converts the bound to KL divergence using stochastic noise. However, this line of argument does not give the right dependence on the dimension for the diffusion setup we study. If one applies their steps directly to our setting, the bound becomes $O(d^{1.5})$ rather than $O(d)$. This point is already shown in detail in lines 378–390 of the original submission (lines 386-398 of the revised with reference added) and now the references to all the three works and a brief part of this discussion has also been added (marked in red) in introduction (lines 83-87, 90-95), related work (lines 146-152).
>
>
> Reaching an $O(d)$ dependence in our paper is not a minor extension. It requires proof ideas that go beyond what exists in the current literature. In particular, we must generalize the stochastic localization argument used in Benton et al. (lines 394–403 of the original submission and lines 402-412 of the revised). Their result only needs control over first order spatial derivatives. In contrast, our analysis must control how the full derivative changes with respect to second order spatial terms (Eq. 12). This structural difference creates technical obstacles that prior proofs do not address.
>
>
> To overcome this, we introduce several new lemmas that do not appear in any earlier work. We first establish, in the diffusion setting, the key property that Benton et al. rely on for the ODE case (Lemma 11). We then extend this to cover expressions involving both the score and its first order derivative (Lemma 12). We further expand the framework to handle the second order spatial derivative term (Lemma 16), which requires an additional set of non-trivial calculations introduced in Lemma 15. These developments are central elements of Section 4.1.1.
>
> These derivations are not easy extensions of existing ones, and genuine new proof techniques are needed. No prior work has shown how to control the high order spatial derivative dependence that is unique to the analysis presented in this paper. Without these new results, it is not possible to obtain the improved $O(d)$ dimension dependence. For this reason, the technical part of Section 4.1.1 provides a substantive advance over the existing literature.

---

> > ### Author Response · Authors · 2025-11-19
> >
> > We believe the clarification above resolves the novelty concern raised in the review. We hope the reviewer can verify the argument and confirm that the issue has been addressed. If there are any remaining questions or additional points the reviewer would like us to clarify, we would be glad to respond.

---

> > > ### Author Response · Authors · 2025-11-21
> > >
> > > We are following up to check whether any additional details would be helpful. We would greatly appreciate it if the reviewer could confirm that the concern has been addressed.

---

> > > > ### Author Response · Authors · 2025-11-23
> > > >
> > > > Since the discussion period is ending soon, we would greatly appreciate it if the reviewer could confirm whether the concern has been addressed. As noted earlier, achieving a linear dependence on the dimension in this setting is highly nontrivial, and the lemmas we provided are, to the best of our knowledge, new to the literature. If any additional details would be helpful, or if the reviewer would like to suggest revisions to the draft, we would be happy to incorporate them.

---

> > ### Comment · Reviewer_mkej · 2025-11-25
> >
> > I think the authors have misunderstood my comments.
> >
> > I was referring to the technique of controlling the KL divergence for the probability flow ODE with injected noise via the L2 square discretization error, which is highlighted and claimed in the abstract. This technique has been commonly used in the literature.
> >
> > I did not imply that the discretization error analysis in this paper is the same to previous works. The works I cited are higher-order methods, whereas this paper focuses on a first-order method, so their analysis techniques are naturally different.
> > Therefore, I do not change my opinion and will keep my score.

---

> ### Author Response · Authors · 2025-11-26
>
> Based on the suggestions, we have updated the abstract which now clearly mentions the point that technique of controlling the KL divergence for the probability flow ODE with injected noise via the L2 square discretization error is similar to previous works and clearly describes the contribution of this work. Also, as discussed above, we have already added this discussion in the main draft along with the discussion that any straightforward adaptation of the techniques used in prior works combined with this KL control formulation cannot lead to the desired dependence on the dimension when considering the discretization error for the ODE (lines 93-98, 150-156, 386-398 of the latest revision).
>
> We belive that this change has resolved all the concerns, and hope the reviewer can acknowledge that the discretization error control for the ODE we have provided in this work is non-trivial and cannot be straighforwardly derived using any of the previous works in the literature, thereby achieving the desired dependence on the KL divergence is a significant contribution.

---

### Meta-Review · Area_Chair_exQM · 2026-01-06

**Summary:**

This paper presents a refined KL-convergence analysis for DDPM sampling under minimal and arguably reasonable assumptions. The central technical contribution is a novel framework for controlling probability-flow ODE discretization error in KL divergence with linear dependence on dimension, improving on the prior best inverse quadratic dependence.

While one reviewer questioned the novelty by focusing on the high-level ODE-plus-noise interpretation, however, my read of things is that the core contribution lies in the new analytic machinery developed to handle second-order spatial derivative terms that arise uniquely in the ODE discretization analysis. The paper introduces novel mathematical techniques to overcome this barrier, enabling the improved dimension dependence.

**Reviewer Concerns:**

Two reviewers found the work technically sound and above the acceptance threshold, and one explicitly described the ideas as “genuinely new.” The remaining negative review maintained a low score despite extensive clarification and revisions, largely due to a disagreement about how novelty should be framed rather than an identified technical flaw.

**Reviewer Scores:**

I don't believe the reviewers would have changed their scores much.

---

### Decision · Program_Chairs · 2026-01-26

Accept (Poster)